# Bridging Past and Future: Distribution-Aware Alignment for Time Series Forecasting

**Yifan Hu**[1,*] **Jie Yang**[3,*]**, Tian Zhou**[2,♠]**, Peiyuan Liu**[1]**, Yujin Tang**[4]**, Rong Jin**[2]**, Liang Sun**[2,†]

[1]Tsinghua University [2]DAMO Academy, Alibaba Group
[3]University of Illinois Chicago [4]Dartmouth College
{huyf0122,yangjie.workmail,peiyuanliu.edu,tangyujin0275}@gmail.com
{tian.zt,liang.sun}@alibaba-inc.com

## Abstract

Although contrastive and other representation-learning methods have long been explored in vision and NLP, their adoption in modern time series forecasters remains limited. We believe they hold strong promise for this domain. To unlock this potential, we explicitly align past and future representations, thereby bridging the distributional gap between input histories and future targets. To this end, we introduce TimeAlign, a lightweight, plug-and-play framework that establishes a new representation paradigm, distinct from contrastive learning, by aligning auxiliary features via a simple reconstruction task and feeding them back into any base forecaster. Extensive experiments across eight benchmarks verify its superior performance. Further studies indicate that the gains arise primarily from correcting frequency mismatches between historical inputs and future outputs. Additionally, we provide two theoretical justifications for how reconstruction improves forecasting generalization and how alignment increases the mutual information between learned representations and predicted targets. Code is in available at https://github.com/TROUBADOUR000/TimeAlign.

## 1 Introduction

Representation learning techniques have been widely explored for time series forecasting (TSF) (Yue et al., 2022; Eldele et al., 2024; Liu & Chen, 2024; Yang et al., 2025a), inspired by their success in vision (Chen & He, 2021; Yu et al., 2025; Yao et al., 2025) and language tasks (Liu et al., 2023b; Dong et al., 2024). They enable forecasters to gain a deeper understanding of the underlying patterns by automatically deriving efficient representations from time series data. However, for the forecasting task, mapping embeddings extracted from history to future distributions is inherently challenging under distribution shift. *Consequently, models often rely on simplistic direct predictive mappings rather than cultivating richer future-aligned representations, leaving the potential of representation learning largely underexploited in modern forecasters.* We argue that this discrepancy is not merely an implementation issue, but rather stems from a fundamental limitation of the prevailing forecasting paradigm as shown in Fig. 2(a). Below, we thoroughly analyze several failure modes in modern forecasting and propose a novel prediction–reconstruction–alignment solution.

❶ *A shortcut through low-frequency periodicity.* We find that models often exploit this by overemphasizing low-frequency periodic components (Huang et al., 2025; Cai et al., 2024). When representation embeddings are optimized solely under error-driven objectives, such as mean squared error (MSE) or mean absolute error (MAE), the resulting predictions tend to oscillate around the learned mean, following dominant periodic patterns (see Fig. 1(a)). Besides, the similarity within the predicted patches is so high that it indicates the forecast is just a repetition of the low-frequency periodic signals extracted from the past (see Fig. 1(b)). Furthermore, as in Fig. 2(c), our empirical analysis across multiple datasets reveals that high-frequency components in the true values are significant, as they reflect abrupt variations beyond simple periodic repetition and are indispensable for robust forecasting. However, these high-frequency signals are often underestimated in the predictions.

---

*Equal contribution ♠ Project lead † Corresponding author

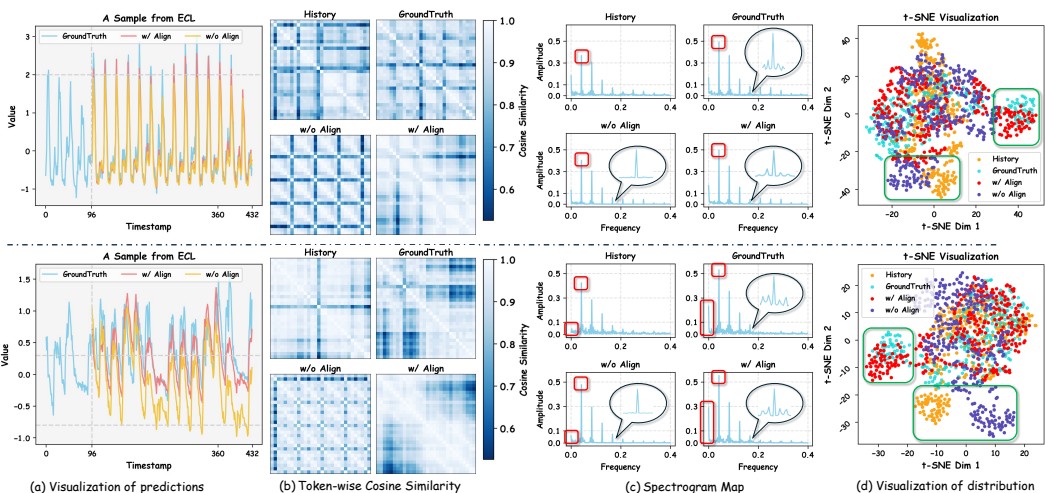

Figure 1: Comparison among history, ground truth, forecast with and without alignment using different visualization perspectives. (a) Time Series Visualization. (b) Token (Patch)-wise Cosine Similarity. (c) Spectrogram Map. (d) t-SNE Embedding Space. More examples are in Appendix F.

❷ *Distributional mismatch between past and future.* As illustrated in Fig. 1(d), the distribution of learned, high-dimensional representations often differs from the distribution of predicted targets (Kim et al., 2022). In practice, these embeddings tend to favour the statistical characteristics of the past over the future. This discrepancy arises from the inherent shift in distribution between historical sequences and the signals to be forecasted (Fan et al., 2023). Consequently, mapping embeddings extracted from history directly to the target distribution becomes challenging (Dai et al., 2024b; Liu et al., 2023a). To overcome this issue, the representation learning process requires additional constraints to ensure that the learned embeddings faithfully capture historical patterns and are also informative for future prediction.

❸ *Structural Flaw of the Unidirectional Paradigm.* Most prevailing prediction paradigm (Fig. 2(a)) is fundamentally unidirectional, mapping historical input solely to the prediction target. This structural constraint inherently limits the model's ability to retain crucial target information. Consequently, as representations are sequentially encoded through deep layers, the process inadvertently acts like a frequency smoother, discarding subtle frequency structures. And these fine-grained details often encode sudden variations triggered by time-varying external events. After being smoothed out, such information cannot be exploited during forecasting, resulting in a consistently low frequency correlation between predictions and ground truth (see Fig. 2(d)). Therefore, modeling fine-grained dynamics effectively is essential for enhancing robustness against abrupt fluctuations.

Collectively, these three observations underscore fundamental limitations of the current paradigm and thus prompt the following question: ***is it possible to design a forecasting framework that can bridge the gap between the past and the future, mitigating distributional shifts while faithfully capturing multi-frequency dynamics?***

To this end, we introduce TimeAlign, which forecasts with an auxiliary reconstruction branch for guidance, as shown in Fig. 2(b), illustrating the prediction–reconstruction–alignment pipeline. The intuition behind reconstruction is twofold. Firstly, unlike forecasting, reconstruction embeddings are naturally aligned with the target distribution since the reconstruction task involves recovering inputs from themselves. Secondly, to faithfully recover signals, reconstruction encourages attention to spectral details, retaining both coarse periodic structures and fine-grained, high-frequency variations. Furthermore, we demonstrate that representations optimized under reconstruction loss exhibit stronger forecasting generalization (see the proof in Appendix B). Consequently, by jointly training the predict and reconstruct branches and aligning their intermediate embeddings, we constrain the predictive representations to be distribution-aware and detail-preserving.

As for alignment, TimeAlign employs two complementary mechanisms. Global alignment reduces the distributional discrepancy between prediction and reconstruction representations, while local alignment enforces fine-grained consistency at the token level. Together, these modules enrich the predictive representation space, enabling the forecaster to capture both periodic low-frequency sig-

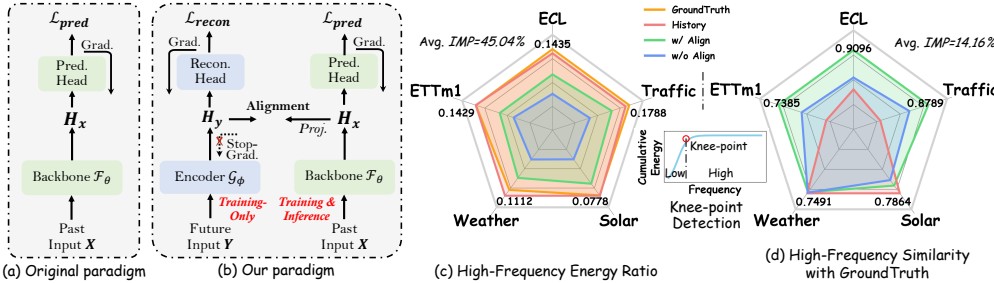

Figure 2: (a) The original paradigm of deep learning forecasters. Distributions are extracted from history and mapped to the prediction space. (b) The paradigm of TimeAlign. Joint optimization of the predict and reconstruct branches provides distributional alignment. (c) High-frequency energy ratio. For different datasets, the threshold between high- and low-frequency bands is determined adaptively via knee-point detection of the cumulative energy distribution. (d) High-frequency similarity. Pearson correlations are computed on the high-frequency components between the ground truth and the history, the forecast with and without alignment. *IMP.* means improvement.

nals and high-frequency dynamics. Importantly, alignment reflects a deeper principle. Predictive representations inherently contain mutual information with future targets, yet conventional objectives underutilize it. Our analyses confirm that TimeAlign acts as an implicit mutual information enhancer, aligning predictive representations more closely with targets (see the proof in Appendix C).

Extensive experiments demonstrate TimeAlign effectively bridges past and future and delivers more reliable forecasts across diverse datasets. In a nutshell, our contributions are summarized as follows:

① ***Paradigm Shift.*** We revisit the representation learning in the current dominant forecasting paradigm and reveal its key limitations. It overemphasizes low-frequency periodicity, suffers from distribution mismatch, and discards informative fine-grained components. To fill these gaps, we introduce a new paradigm incorporating alignment between prediction and reconstruction.

② ***Technical Contribution.*** We present TimeAlign, a dual-branch framework that consists of a predict branch and a reconstruct branch, distribution-aware aligning their latent representations through global and local objectives to bridge the past and future effectively.

③ ***In Theory and In Practice.*** We provide two theoretical guarantees: (i) reconstruction improves the forecasting generalization, and (ii) alignment enhances the mutual information between predictive representations and future targets. Extensive experiments across diverse benchmarks further validate these insights, establishing state-of-the-art performance in both accuracy and robustness.

## 2 RELATED WORKS

### 2.1 DEEP LEARNING IN TIME SERIES FORECASTING

Recently, deep learning has become the dominant paradigm for TSF, with a variety of neural architectures including CNN (Donghao & Xue, 2024; Wu et al., 2023; Wang et al., 2022), Linear (Huang et al., 2025; Hu et al., 2025c; Si et al., 2025), Transformer (Liu et al., 2024b; 2025; Dai et al., 2024a), GNN (Hu et al., 2025d; Wu et al., 2020), etc. Despite architectural diversity, these methods generally follow an encoder–decoder paradigm: they embed historical sequences into latent representations and map them to predictive targets (see Fig. 2(a)). However, as shown in Fig. 1, this paradigm tends to overemphasize low-frequency patterns while neglecting fine-grained structures, and it suffers from distribution mismatch between history and future. To address these limitations, we propose TimeAlign, which jointly optimizes predict and reconstruct branches and aligns their embeddings, thereby bridging the gap between the past and the future more effectively.

### 2.2 REPRESENTATION LEARNING

Representation learning is crucial for enabling models to capture meaningful and transferable features. Across domains, it has been used to introduce inductive biases that extend beyond task-specific objectives. For instance, in diffusion models, learned representations serve as structured noise priors that improve sample quality and stability (Yang et al., 2025c; Yu et al., 2025; Jiang et al., 2025), while in vision and language, aligning task-specific embeddings with pre-trained ones

enriches feature spaces and boosts downstream performance (Yao et al., 2025; Leng et al., 2025). However, in TSF, error-driven training often yields biased embeddings that smooth away subtle but critical dynamics Yue et al. (2022); Eldele et al. (2024), limiting their predictive value. We argue that auxiliary mechanisms are needed to enforce consistency between predictive and reconstructive signals, thereby preserving fine-grained temporal cues and alleviating distribution mismatch.

## 3 METHOD

### 3.1 PROBLEM DEFINITION

In TSF task, let $\mathbf{X} = \{x_1, x_2, ..., x_C\} \in \mathbb{R}^{C \times L}$ be the history input series, where $C$ denotes the number of channels and $L$ denotes the look-back horizon. $\mathbf{x}_i \in \mathbb{R}^L$ represents one of the channels. The objective is to construct a model $\mathcal{F}_\theta(\cdot)$ that predicts the future sequences $\hat{\mathbf{Y}}_{\text{pred}} \in \mathbb{R}^{C \times T}$, where $T$ denotes the forecasting horizon. The forecasting process is given by $\hat{\mathbf{Y}}_{\text{pred}} = \mathcal{F}_\theta(\mathbf{X})$. A lower error between $\hat{\mathbf{Y}}_{\text{pred}}$ and ground truth $\mathbf{Y}$ indicates stronger forecasting capability of the model $\mathcal{F}_\theta(\cdot)$. However, $\mathbf{X}$ and $\mathbf{Y}$ often exhibit distribution shift. Instead of predicting patterns in $\mathbf{Y}$ using learnable statistics derived from $\mathbf{X}$, our focus is to leverage direct constraint from $\mathbf{Y}$ to guide model optimization, thereby achieving robust and efficient distribution alignment between $\hat{\mathbf{Y}}_{\text{pred}}$ and $\mathbf{Y}$. To this end, we introduce an auxiliary reconstruction task. Specifically, the reconstruction model is defined as $\mathcal{G}_\phi(\cdot)$ that directly maps $\mathbf{Y}$ to its reconstruction $\hat{\mathbf{Y}}_{\text{recon}} = \mathcal{G}_\phi(\mathbf{Y})$. Here, $\mathcal{G}_\phi(\cdot)$ learns a compact representation of the target distribution without relying on past inputs, and the reconstruct representation $\mathbf{H}_y$ serves as a guidance for aligning the latent space $\mathbf{H}_x$ of $\mathcal{F}_\theta(\cdot)$.

### 3.2 STRUCTURE OVERVIEW

As illustrated in Fig. 3, our proposed TimeAlign consists of four components: (i) **Predict Branch** employs a backbone encoder to map $\mathbf{X}$ to $\hat{\mathbf{Y}}_{\text{pred}}$, active in both training and inference. The backbone is *plug-and-play* and replaceable by any forecasting architecture. In our default setup, it shares the same lightweight encoder as the Reconstruct Branch to highlight the effect of alignment. (ii) **Reconstruct Branch** leverages a lightweight encoder to reconstruct $\hat{\mathbf{Y}}recon$ from $\mathbf{Y}$, thereby capturing a compact representation aligned with the target distribution. This branch is used only during training. (iii) **Distribution-Aware Alignment** explicitly aligns $\mathbf{H}_x$ and $\mathbf{H}_y$ via global and local mechanisms, providing distributional constraint for optimization. (iv) **A Simple Encoder** serves as the minimal yet effective design for the Reconstruct Branch and default Predict Branch.

### 3.3 PREDICT BRANCH

The predict branch follows the standard deep learning paradigm for TSF, extracting temporal representations from historical input and mapping them into the future horizon. In TimeAlign, we deliberately adopt a minimal encoder design to demonstrate that the improvements stem from alignment rather than architectural complexity. Importantly, the encoder here can be replaced by more sophisticated backbones, enabling TimeAlign to serve as a plug-and-play module.

Given the history input $\mathbf{X} \in \mathbb{R}^{C \times L}$, we first divide it into patches and project them onto embeddings $\mathbf{X}_p = \text{Linear}(\text{Patching}(\mathbf{X})) \in \mathbb{R}^{(C \cdot N) \times D}$, where $N$ is the number of patches and the Reconstruct Branch share the same $N$ with the Predict Branch. $\mathbf{X}_p$ is then processed by an $M$-layer encoder, where each layer consists of a linear–activation–linear block with residual connection:

$$\mathbf{H}_x^l = \mathbf{H}_x^{l-1} + \text{Linear}\left(\sigma\left(\text{Linear}(\mathbf{H}_x^{l-1})\right)\right), \quad l = 1, \dots, M \tag{1}$$

with $\mathbf{H}_x^0 = \mathbf{X}_p$ and $\sigma$ is an activation function. Finally, a prediction head projects the latent representation into the forecasting results $\hat{\mathbf{Y}}_{\text{pred}} = \text{Linear}(\mathbf{H}_x^M)$.

### 3.4 RECONSTRUCT BRANCH

The reconstruct branch provides explicit constraint from the target distribution by reconstructing the ground truth from itself. This design captures the intrinsic structure of $\mathbf{Y}$, thereby offering a reliable

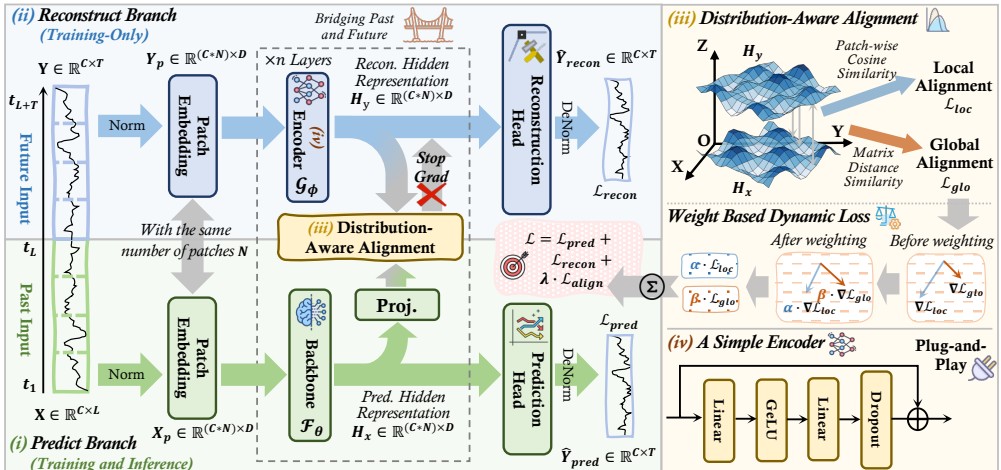

Figure 3: Overall architecture of TimeAlign. (i) **Predict Branch** maps history to forecasts (both training and inference), with a *replaceable* backbone. (ii) **Reconstruct Branch** reconstructs targets to capture the distribution (training-only). (iii) **Distribution-Aware Alignment** aligns predict and reconstruct representations via global and local mechanisms. (iv) **A Simple Encoder** is the default lightweight design in Reconstruct Branch and default Predict Branch.

distributional $\mathbf{H}_y$ reference for alignment. Unlike the predict branch, the reconstruct branch is only active during training and discarded at inference.

Similarly, given the future sequence $\mathbf{Y} \in \mathbb{R}^{C \times T}$, we embed it in the same way of patch embedding $\mathbf{Y}_p = \text{Linear}(\text{Patching}(\mathbf{Y})) \in \mathbb{R}^{(C \cdot N) \times D}$. It is then encoded by the same $M$-layer encoder:

$$\mathbf{H}_y^l = \mathbf{H}_y^{l-1} + \text{Linear}\left(\sigma\left(\text{Linear}(\mathbf{H}_y^{l-1})\right)\right), \quad l = 1, \ldots, M \tag{2}$$

with $\mathbf{H}_y^0 = \mathbf{Y}_p$. Finally, a reconstruction head then maps the latent representation back into the target space $\hat{\mathbf{Y}}_{\text{recon}} = \text{Linear}(\mathbf{H}_y^M)$.

The hidden representations $H_x$ and $H_y$ are structured by deep transformations and normalization layers. In the spirit of the Central Limit Theorem (CLT), these compounding effects typically enforce the distribution of $H_x$ and $H_y$ to be approximately Gaussian. This structured property is key to the theoretical generalization advantages discussed in Appendix B.

## 3.5 DISTRIBUTION-AWARE ALIGNMENT

Although the predict and reconstruct branches encode their respective inputs independently, the latent spaces still diverge due to distributional shift between the history $\mathbf{X}$ and the target $\mathbf{Y}$. To explicitly bridge this gap, we introduce a distribution-aware alignment module that aligns the representations of the two branches at each layer. We consider the $i$-th layer as an example below.

**Alignment with an Additional Mapping.** Instead of aligning the raw hidden states directly, we introduce an additional linear mapping applied to the predict branch before alignment:

$$\tilde{\mathbf{H}}_x^i = \text{Linear}(\mathbf{H}_x^i) \in \mathbb{R}^{(C \cdot N) \times D}. \tag{3}$$

This mapping layer, implemented as a lightweight Linear layer, serves to temporally reposition and normalize the highly abstracted features of the predict branch, making them more compatible with the representations from the reconstruct branch. Importantly, gradients from the alignment loss are only propagated to the predict branch via this Linear layer, leaving the reconstruct branch unaffected (Chen & He, 2021). This asymmetric gradient flow ensures that the reconstruct branch provides a stable supervisory signal without being distorted by the alignment procedure.

**Local vs. Global Complementarity.** To ensure fine-grained fidelity and distributional consistency, we introduce two complementary objectives, including local alignment to capture patch-wise similarity and global alignment to capture relational structure (Yao et al., 2025; Yang et al., 2025b).

Local alignment ensures that individual patch-level features capture similar semantics, while global alignment enforces consistency in the overall relational structure of representations (Yang et al., 2026). Specifically, local alignment preserves fine-grained similarity between corresponding patches, which is crucial for capturing sharp transitions and high-frequency details. Notably, such similarity measures spectral energy projection, thereby leveraging multi-frequency information. $h_{x,j}^i$ is the $j$-th item in $\mathbf{H}_x^i$ and $n = C \cdot NL$:

$$\mathcal{L}_{\text{local}}^i = \frac{1}{n^2} \sum_{j=1}^{n} \text{GELU} \left( 1 - \left| \tilde{h}_{x,j}^i \cdot h_{y,j}^i \right| - \delta_{loc} \right). \tag{4}$$

Meanwhile, global alignment aims to make the relative distance matrices within the features as similar as possible, ensuring that large-scale temporal dependencies and low-frequency dynamics are also aligned. $\top$ means the transpose. We subtract margins $\delta_{loc}$ and $\delta_{glo}$ to relax the constraint:

$$\mathcal{L}_{\text{global}}^i = \frac{1}{n^2} \sum_{j=1}^{n} \text{GELU}(\left\| \tilde{h}_{x,j}^i (\tilde{h}_{x,j}^i)^\top - h_{y,j}^i (h_{y,j}^i)^\top \right\|_1 - \delta_{glo}). \tag{5}$$

These margins are used to relax the constraint, which is a common technique in metric learning. They create a "buffer" where, if two representations are already close enough, the loss becomes zero. This prevents the model from overfitting on easy samples and focuses its optimization effort on harder-to-align pairs, leading to more stable training.

**Weight-Based Dynamic Loss.** Finally, the overall alignment objective is a weighted fusion

$$\mathcal{L}_{\text{align}} = \frac{1}{M} \sum_{i=1}^{M} \left( \alpha \mathcal{L}_{\text{local}}^i + \beta \mathcal{L}_{\text{global}}^i \right). \tag{6}$$

We adopt an adaptive scheme to prevent domination of either objective and stabilizes optimization across different datasets and regimes. The dynamic weights $\alpha$ and $\beta$ are computed as

$$\alpha = \frac{\mathcal{L}_{\text{local}} + \mathcal{L}_{\text{global}}}{\mathcal{L}_{\text{local}}}, \quad \beta = \frac{\mathcal{L}_{\text{local}} + \mathcal{L}_{\text{global}}}{\mathcal{L}_{\text{global}}}. \tag{7}$$

Combining local and global alignment under an asymmetric mapping design encourages the predict branch to capture detailed, patch-level similarities while respecting global relational structures. This dual alignment, guided by the stable target distribution from the reconstruct branch, enhances the robustness and fidelity of the forecasting process.

### 3.6 Optimization Objective

The full optimization combines the prediction loss, reconstruction loss, and alignment loss:

$$\mathcal{L} = \mathcal{L}_{\text{pred}} + \mathcal{L}_{\text{recon}} + \lambda \mathcal{L}_{\text{align}} \tag{8}$$

where $\mathcal{L}_{\text{pred}}$ and $\mathcal{L}_{\text{recon}}$ are error-based MAE objectives and $\lambda$ is a scaling weight. This joint training paradigm enables the predict branch to benefit directly from distributional constraint provided by the reconstruct branch.

## 4 Theoretical Analysis

### 4.1 Improve generalization with reconstruction error minimization

The following theorem characterizes the advantage of reconstruction-guided estimation. Let $\mathcal{M}^*$ be the optimal forecasting solution, $\mathcal{M}_1$ the empirical estimator obtained from the training examples, and $\mathcal{M}_2$ the estimator derived from the one-dimensional representation via reconstruction error minimization. Given the concentration bounds, the reconstruction-based estimator $\mathcal{M}_2$ achieves a strictly tighter approximation to $\mathcal{M}^*$ than $\mathcal{M}_1$. The proof can be found in Appendix B.

$$\frac{|\mathcal{M}_2 - \mathcal{M}_*|_F}{|\mathcal{M}_*|_F} \ll \frac{|\mathcal{M}_1 - \mathcal{M}_*|_F}{|\mathcal{M}_*|_F} \tag{9}$$

where $|\cdot|_F$ denotes the Frobenius norm.

## 4.2 TimeAlign is an implicit mutual information enhancer

To understand the mechanism of $\mathcal{L}_{\text{align}}$ from a structural perspective, we reformulate the role of our dual loss objectives through the lens of Mutual Information Maximization (MIM). The conventional prediction loss, $\mathcal{L}_{\text{pred}}$, provides a variational lower bound on the mutual information $I(\mathbf{Y}; \mathbf{H}_x)$, implicitly guiding the representation learning. Our core alignment of TimeAlign acts as a complementary mechanism, ensuring that the predictive representation $\mathbf{H}_x(h_x)$ is explicitly enriched by maximizing its mutual information with the ground truth $\mathbf{Y}(y)$5. This effectively enhances the total information flow used for forecasting. The mutual information is formulated as:

$$I(\mathbf{Y}; \mathbf{H}_x) = \mathbb{E}_{p(y,h_x)} \left[ \log \frac{p(y, h_x)}{p(h_x) \cdot p(y)} \right]. \tag{10}$$

**MIM in Prediction Loss.** Following the derivations introduced in previous IB-relevant works (Choi & Lee, 2023; Alemi et al., 2016), we can obtain a lower bound for the informative term, which aims to maximize the mutual information between $\mathbf{H}_x$ and $\mathbf{Y}$ (see full derivation in Appendix C.1):

$$\begin{aligned} I(\mathbf{Y}; \mathbf{H}_x) &= \mathbb{E}_{p(y,h_x)} \left[ \log \frac{\mathcal{F}_\theta(y|h_x)}{p(y)} \right] + \int_{h_x} p(h_x) \cdot D_{\text{KL}}[p(y|h_x)||\mathcal{F}_\theta(y|h_x)] \, dh_x \\ &\geq \mathbb{E}_{p(y,h_x)} \left[ \log \mathcal{F}_\theta(y|h_x) \right] \overset{\text{def}}{=} -\mathcal{L}_{\text{pred}}, \end{aligned} \tag{11}$$

where the inequality holds due to the non-negativity of the KL divergence and entropy.

**MIM in Alignment Loss.** To further enhance the mutual information, we introduce a complementary formulation that explicitly targets the mutual information between $\mathbf{H}_x$ and $\mathbf{Y}$. Inspired by the InfoNCE objective from contrastive learning (Oord et al., 2018), we derive an alternative lower bound of $I(\mathbf{Y}; \mathbf{H}_x)$ (see the full derivation in Appendix C.2). Furthermore, instead of predicting $y$ directly, we model a density ratio $f(y, h_x) = \exp(\text{proj}(h_x)^\top \cdot \mathcal{G}_\phi(y))$ that preserves mutual information $I(Y; Z)$, as it is proportional to $\frac{p(y|z)}{p(y)}$. This yields the following Alignment loss.

$$\begin{aligned} I(\mathbf{Y}; \mathbf{H}_x) &= -\mathbb{E}_{p(y,h_x)} \left[ \log \left( \frac{p(y)}{p(y|h_x)} \cdot N \right) - \log N \right] \approx -\mathbb{E}_{p(y,h_x)} \left[ \log \left( \frac{p(y)}{p(y|h_x)} \cdot N \right) \right] \\ &= -\mathbb{E}_{p(y,h_x)} \left[ \exp \left( \text{proj}(h_x)^\top \cdot \mathcal{G}_\phi(y) \right) \right] \overset{\text{def}}{=} -\mathcal{L}_{\text{align}}. \end{aligned} \tag{12}$$

In summary, jointly optimizing $\mathcal{L}_{\text{pred}}$ and $\mathcal{L}_{\text{align}}$ effectively maximizes the mutual information $I(\mathbf{Y}; \mathbf{H}_x)$, thereby enabling more accurate and robust representation learning. To further refine $\mathcal{L}_{\text{align}}$, we adopt the complementary local and global strategies discussed in Sec. 3.5, which together ensure both fine-grained fidelity and distributional consistency.

## 5 Experiment

In this section, we conduct extensive experiments to validate the superiority of our TimeAlign in **effective forecasting performance**, **plug-and-play capability**, and **competitive efficiency**. More experimental details can be found in Appendix D and Appendix E.

### 5.1 Experiment Setup

**Datasets.** We conduct experiments on eight widely used real-world datasets, including the Electricity Transformer Temperature (ETT) dataset with its four subsets (ETTh1, ETTh2, ETTm1, ETTm2), as well as Weather, Electricity, Traffic, and Solar (Miao et al., 2025; 2024). The statistics of the datasets are shown in Appendix D.2.

**Baselines.** We compare TimeAlign with several SOTA baselines representing the latest advances in TSF, including TimeMixer (Wang et al., 2024), iTransformer (Liu et al., 2024b), PatchTST (Nie et al., 2023), Crossformer (Zhang & Yan, 2023), TVNet (Li et al., 2025), ModernTCN (Donghao & Xue, 2024), TimesNet (Wu et al., 2023), and lightweight methods like DLinear (Zeng et al., 2023), CMoS (Si et al., 2025), TimeBase (Huang et al., 2025) and Leddam (Yu et al., 2024).

| Models | TimeAlign (Ours) | | CMoS (2025) | | TimeBase (2025) | | TVNet (2025) | | iTransformer (2024b) | | TimeMixer (2024) | | Leddam (2024) | | ModernTCN (2024) | | PatchTST (2023) | | Crossformer (2023) | | TimesNet (2023) | | DLinear (2023) | |
|---|---|---|---|---|---|---|---|---|---|---|---|---|---|---|---|---|---|---|---|---|---|---|---|---|
| Metric | MSE | MAE | MSE | MAE | MSE | MAE | MSE | MAE | MSE | MAE | MSE | MAE | MSE | MAE | MSE | MAE | MSE | MAE | MSE | MAE | MSE | MAE | MSE | MAE |
| ETTm1 | 0.340 | 0.367 | 0.354 | 0.378 | 0.357 | 0.380 | 0.348 | 0.379 | 0.362 | 0.391 | 0.355 | 0.380 | 0.354 | 0.381 | 0.351 | 0.381 | 0.353 | 0.382 | 0.420 | 0.435 | 0.400 | 0.406 | 0.356 | 0.378 |
| ETTm2 | 0.243 | 0.302 | 0.259 | 0.316 | 0.250 | 0.314 | 0.251 | 0.311 | 0.269 | 0.329 | 0.257 | 0.318 | 0.265 | 0.320 | 0.253 | 0.314 | 0.256 | 0.317 | 0.518 | 0.501 | 0.292 | 0.330 | 0.259 | 0.324 |
| ETTh1 | 0.406 | 0.420 | 0.403 | 0.416 | 0.396 | 0.414 | 0.407 | 0.421 | 0.439 | 0.448 | 0.427 | 0.441 | 0.415 | 0.430 | 0.404 | 0.420 | 0.418 | 0.436 | 0.440 | 0.463 | 0.458 | 0.450 | 0.424 | 0.439 |
| ETTh2 | 0.336 | 0.382 | 0.331 | 0.383 | 0.345 | 0.397 | 0.324 | 0.377 | 0.374 | 0.406 | 0.349 | 0.397 | 0.345 | 0.391 | 0.322 | 0.379 | 0.351 | 0.404 | 0.809 | 0.658 | 0.390 | 0.427 | 0.431 | 0.447 |
| Weather | 0.214 | 0.244 | 0.220 | 0.261 | 0.219 | 0.263 | 0.221 | 0.261 | 0.233 | 0.271 | 0.226 | 0.264 | 0.226 | 0.264 | 0.224 | 0.264 | 0.226 | 0.264 | 0.228 | 0.287 | 0.259 | 0.287 | 0.242 | 0.293 |
| Electricity | 0.154 | 0.244 | 0.158 | 0.250 | 0.167 | 0.258 | 0.165 | 0.254 | 0.164 | 0.261 | 0.185 | 0.284 | 0.162 | 0.256 | 0.156 | 0.253 | 0.159 | 0.253 | 0.181 | 0.279 | 0.192 | 0.295 | 0.166 | 0.264 |
| Traffic | 0.378 | 0.240 | 0.396 | 0.270 | 0.417 | 0.278 | 0.396 | 0.268 | 0.397 | 0.282 | 0.409 | 0.279 | 0.452 | 0.283 | 0.396 | 0.270 | 0.397 | 0.275 | 0.523 | 0.284 | 0.620 | 0.336 | 0.418 | 0.287 |
| Solar | 0.192 | 0.214 | 0.210 | 0.257 | 0.236 | 0.270 | 0.228 | 0.277 | 0.202 | 0.248 | 0.193 | 0.264 | 0.223 | 0.264 | 0.228 | 0.281 | 0.194 | 0.245 | 0.205 | 0.232 | 0.244 | 0.334 | 0.224 | 0.226 |

Table 1: Long-term forecasting results. The input length $L$ is searched from $\{336, 512, 720\}$ and the results are averaged across four forecasting horizons $T \in \{96, 192, 336, 720\}$. The best and second-best results are highlighted in **bold** and underlined, respectively. See Tab. 6 for full results.

| Case | Local Align w/ | Global Align w/ | ETTm1 MSE | MAE | ETTm2 MSE | MAE | Weather MSE | MAE | Electricity MSE | MAE | Traffic MSE | MAE |
|---|---|---|---|---|---|---|---|---|---|---|---|---|
| ① | × | × | 0.349 | 0.370 | 0.252 | 0.308 | 0.225 | 0.254 | 0.159 | 0.248 | 0.390 | 0.249 |
| ② | ✓ | × | 0.344 | 0.372 | 0.245 | 0.305 | 0.220 | 0.249 | 0.157 | 0.247 | 0.383 | 0.244 |
| ③ | × | ✓ | 0.342 | 0.369 | 0.244 | 0.303 | 0.218 | 0.247 | 0.157 | 0.246 | 0.383 | 0.245 |
| ④ | ✓ | ✓ | 0.340 | 0.367 | 0.243 | 0.302 | 0.214 | 0.244 | 0.154 | 0.244 | 0.378 | 0.240 |

Table 2: Full results of ablation on the effect of removing alignment in local and global perspective. ✓ indicates the use of alignment, while × means alignment is retained.

**Implementation Details.** All experiments are performed on one NVIDIA V100 32GB GPU. We select two common metrics in TSF: MAE and MSE. Moreover, a grid search is conducted on the look-back length of $\{96, 192, 336, 512, 720\}$. For each setting, the results are averaged over three runs with random seeds. For additional details on hyperparameters, please refer to Appendix D.3.

## 5.2 Forecsating Performance

As shown in Tab. 1, TimeAlign delivers **effective forecasting performance** across most datasets. The only exceptions are ETTh1 and ETTh2, yet its performance remains highly competitive. Most notably, on the ETTm1 and ETTm2 datasets known for severe distribution shifts (Qiu et al., 2024), TimeAlign substantially outperforms current SOTA methods. Owing to the Distribution-Aware Alignment Module, TimeAlign aligns predict representations with evolving distributions, thereby enabling more detailed modeling of the target and further boosting accuracy. Overall, compared to the second best method TVNet, TimeAlign reduces MSE/MAE by $3.27\%/5.20\%$, with a Wilcoxon test p-value of $1.37e^{-8}$ confirming significance at $99\%$ confidence. The full error bars are in Tab. 7.

## 5.3 Ablation Studies

We validate the effectiveness of each architectural design in TimeAlign through the following ablation studies: ① **w/o Align** signifies the Predict Branch alone without alignment. ② **w/ Local only** denotes the Local Alignment is retained. ③ **w/ Global only** denotes only the Global Alignment is retained. ④ **w/ Align** encompasses both Local and Global Alignment mechanisms.

The results in Tab. 2 reveal that the combination of Local and Global Alignment is essential for accurately reconciling the predict representations with the target distribution, yielding the best performance. Interestingly, Global Alignment alone outperforms Local Alignment alone; the former provides a coarse-grained distribution-level pull that is more effective at high-level representation calibration, while the latter focuses on fine-grained, point-wise corrections. As expected, removing both alignment strategies results in the worst accuracy. Furthermore, we observe that w/o Align is already a SOTA-level Baseline, showing that TimeAlign achieves consistent performance improvements on top of the already-SOTA-level baseline. This proves that our TimeAlign module is not a trick that only works on a weak baseline. It is a powerful module that can take a strong, efficient, SOTA-level architecture and make it even better.

Additionally, the effectiveness of Reconstruct can be found in Appendix E.5.

| Models | iTransformer | | | +TimeAlign | | | | DLinear | | | +TimeAlign | | | |
|---|---|---|---|---|---|---|---|---|---|---|---|---|---|---|
| Metric | MSE | MAE | SIM | MSE | MAE | SIM | $\Delta IMP$ | MSE | MAE | SIM | MSE | MAE | SIM | $\Delta IMP$ |
| ETTm1 | 0.362 | 0.391 | 0.862 | 0.355 | 0.384 | 0.865 | +1.80% | 0.356 | 0.378 | 0.869 | 0.352 | 0.373 | 0.872 | +1.05% |
| ETTm2 | 0.269 | 0.329 | 0.970 | 0.260 | 0.317 | 0.974 | +3.44% | 0.259 | 0.324 | 0.973 | 0.252 | 0.311 | 0.976 | +2.61% |
| ETTh1 | 0.439 | 0.448 | 0.825 | 0.428 | 0.448 | 0.828 | +2.45% | 0.424 | 0.439 | 0.841 | 0.414 | 0.433 | 0.840 | +2.41% |
| ETTh2 | 0.374 | 0.406 | 0.958 | 0.363 | 0.402 | 0.962 | +2.87% | 0.431 | 0.447 | 0.960 | 0.418 | 0.431 | 0.960 | +3.07% |
| Weather | 0.233 | 0.271 | 0.777 | 0.240 | 0.274 | 0.777 | −3.00% | 0.242 | 0.293 | 0.778 | 0.238 | 0.280 | 0.775 | +1.55% |
| Electricity | 0.164 | 0.261 | 0.914 | 0.163 | 0.255 | 0.916 | +0.76% | 0.166 | 0.264 | 0.912 | 0.164 | 0.260 | 0.916 | +1.20% |
| Traffic | 0.397 | 0.282 | 0.845 | 0.390 | 0.260 | 0.848 | +1.95% | 0.418 | 0.287 | 0.832 | 0.418 | 0.285 | 0.833 | +0.12% |
| Solar | 0.202 | 0.248 | 0.852 | 0.204 | 0.243 | 0.853 | −0.87% | 0.224 | 0.226 | 0.848 | 0.214 | 0.219 | 0.852 | +4.15% |

Table 3: Full results of long-term forecasting with TimeAlign as a plugin. All results are averaged across four different prediction lengths: $O \in \{96, 192, 336, 720\}$. $\Delta IMP$ denotes TimeAlign's performance gain, either *positive* or *negative*.

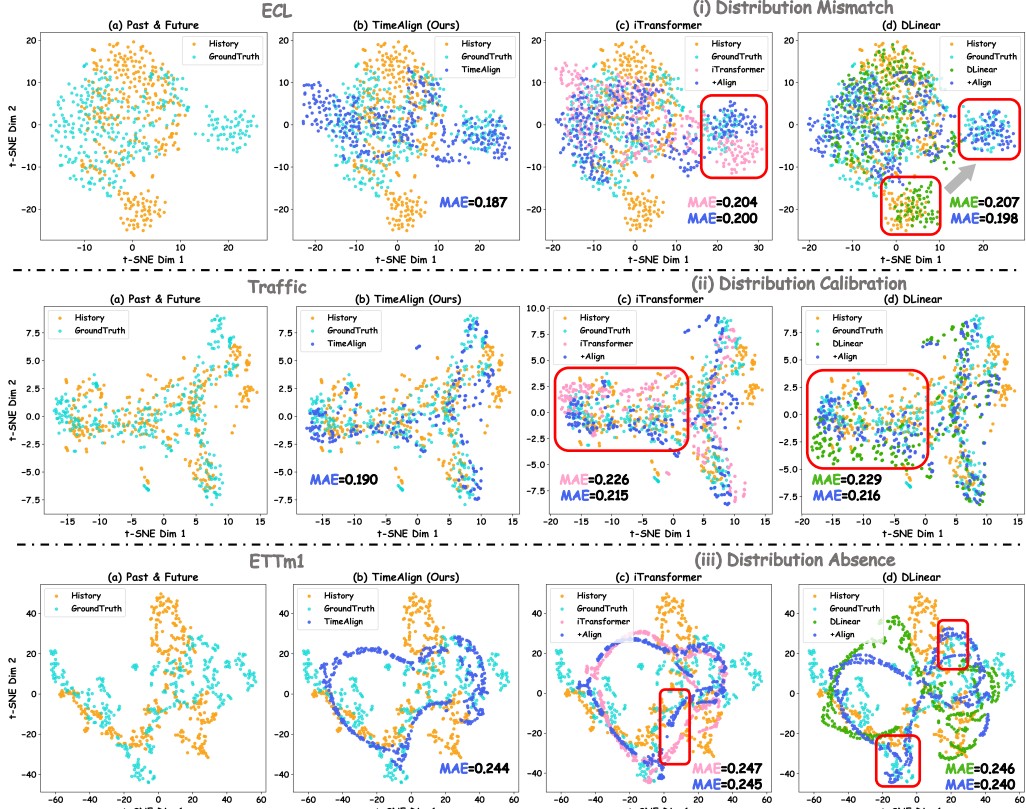

Figure 4: The t-SNE visualization illustrates the distribution of the history, the ground truth and the forecasts produced by TimeAlign, iTransformer (Liu et al., 2024b), DLinear (Zeng et al., 2023), iTransformer+TimeAlign and DLinear+TimeAlign on the ECL, Traffic, and ETTm1 datasets. The TimeAlign forecasts almost perfectly overlap with the ground truth manifold, whereas the predictions from vanilla iTransformer and DLinear exhibit obvious distributional divergence. Plugging in TimeAlign visibly collapses this gap, steering backbones toward the target distribution.

## 5.4 PLUG-AND-PLAY EXPERIMENT

TimeAlign can be seamlessly integrated into diverse forecasting backbones. For encoder-based models, the encoder first maps the history $X$ into a latent embedding $X'$ via patch embedding (Nie et al., 2023), value-wise convolution (Liu et al., 2024b; Wu et al., 2021), or linear projection (Huang et al., 2025). A parallel reconstruct branch with the same architecture projects the target $Y$ into the same space $Y'$, enabling alignment between prediction and reconstruction representations. This guides the backbone beyond point-wise error minimization toward representations more informative of the target. To verify the **plug-and-play capability**, we augment both iTransformer (Liu et al., 2024b) and DLinear (Zeng et al., 2023) and further quantify distributional discrepancy using cosine

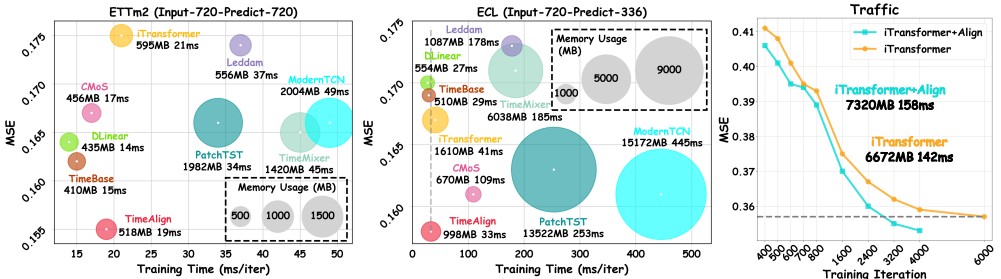

Figure 5: *Left*: Model efficiency comparison under ETTm2 and ECL datasets. *Right*: Training iteration vs. MSE plot. Model training becomes more efficient and effective.

similarity (SIM), and ensure a fair comparison by searching the input length $L$ as Sec. 5.1. As shown in Tab. 3, TimeAlign consistently improves forecasting accuracy by $1\%$–$4\%$ across most benchmarks, while increasing cosine similarity between predictions and ground truth. This dual benefit of low error and high distributional similarity enhances forecast reliability. Performance dips on Weather and Solar datasets are caused by extreme outliers and zero-imputed values, which distort the learned distribution, inflate alignment loss and impair local alignment, respectively. Moreover, these excessively complex architectures hinder the alignment module from effectively correcting the hierarchically extracted representations, leading to inferior performance compared to TimeAlign.

Furthermore, Fig. 5 (right) visualizes the training trajectories of the vanilla and TimeAlign-augmented iTransformer. With the plug-in, TimeAlign guides the encoder towards distributionally coherent minima, slightly increasing GPU memory usage, preserving the original per-iteration latency, and greatly accelerating convergence. On the Traffic dataset, the augmented one reaches the same validation loss 3,000 iterations earlier than the vanilla iTransformer.

## 5.5 ANALYSIS OF DISTRIBUTION-AWARE ALIGNMENT

Fig. 4 presents t-SNE visualizations of the distributions of the history $\mathbf{X}$, ground truth $\mathbf{Y}$, and forecasts from various methods on ECL, Traffic, and ETTm1. (a) $\mathbf{X}$ and $\mathbf{Y}$ exhibit a clear distribution shift with partial overlap. (b) Our $\hat{\mathbf{Y}}_{pred}$ bridge this gap, matching $\mathbf{Y}$'s range and fine-grained structure, including peripheral outlier clusters, highlighting the effectiveness of TimeAlign. (c) and (d) contrast the vanilla and TimeAlign-augmented iTransformer and DLinear, respectively.

Furthermore, we categorize the discrepancies between $\hat{\mathbf{Y}}$ and $\mathbf{Y}$ into three patterns. (i) Distribution Mismatch: $\hat{\mathbf{Y}}$ resembles $\mathbf{X}$ more than $\mathbf{Y}$; (ii) Distribution Calibration: global shapes align but local modes diverge; (iii) Distribution Absence: $\hat{\mathbf{Y}}$ misses some modes of $\mathbf{Y}$. Incorporating TimeAlign markedly reduces all three artifacts. On ECL, DLinear's forecasts shift from $X$'s skew to $Y$'s manifold; on Traffic, both iTransformer's and DLinear's predictions recenter around $Y$'s true mean; on ETTm1, absent regions are recovered, yielding almost identical distributions of $\mathbf{Y}$. These qualitative results confirm TimeAlign is a robust plug-in for distribution alignment across backbones.

## 5.6 EFFECTIVENESS AND EFFICIENCY

Another key advantage of TimeAlign lies in its **competitive efficiency**, delivering superior performance with few time cost. Using official configurations and the same batch size, Fig. 5 shows that as a purely Linear architecture, TimeAlign surpasses all Transformer- and CNN-based methods in both speed and memory usage, while achieving SOTA training times among Linear-based models.

## 6 CONCLUSION

This paper introduces TimeAlign, a distribution-aware alignment framework for time series forecasting. By coupling prediction and reconstruction with global and local alignment, TimeAlign effectively bridges the gap between past and future, preserving critical frequency components and boosting forecasting accuracy. Theoretical evidence demonstrates that reconstruction improves forecasting generalization, while alignment increases the mutual information between learned representations and predicted targets. Extensive experiments validate the effectiveness of TimeAlign across diverse benchmarks. We believe that our findings open up promising avenues for reconsidering representation learning in time series forecasting.

## 7 ETHICS STATEMENT

As our work only focuses on the time series forecasting problem, there is no potential ethical risk.

## 8 REPRODUCIBILITY STATEMENT

In the main text, we have formally defined the model architecture with equations. All the implementation details, including dataset descriptions, metrics, and experiment configurations, are provided in the manuscript.

## ACKNOWLEDGMENTS

This work was supported by DAMO Academy through DAMO Academy Research Intern Program.

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

## A LLM USAGE

We used Large Language Models (LLMs) as auxiliary tools to assist with the writing process. They were used solely to polish the language and improve readability, with no influence over the research design, experimental implementation or analysis. We conceived and executed all methodological contributions, experiments, and conclusions independently.

## B IMPROVE GENERALIZATION WITH RECONSTRUCTION ERROR MINIMIZATION

In this section, we discuss how the representation derived from the minimization of reconstruction error can be used to improve the generalization error. Let $(x_i, y_i), i = 1, \ldots, n$ be the set of training examples, where the **latent** input pattern $x_i \in \mathbb{R}^d$ sampled from an Gaussian distribution $\mathcal{N}(0, I_d)$ and the output vector $y_i = (y_{i,1}, \ldots, y_{i,d}) \in \mathbb{R}^d$ is constructed as follows:

$$y_{i,j} = \begin{cases} w^\top x_i & j = 1 \\ \sim \mathcal{N}(0, \sigma^2) & j > 1 \end{cases} \tag{13}$$

We now consider learning a linear regression model to map the input $x$ to output $y$. It is clear that the optimal solution is given by

$$\mathcal{M}_* = e_1 w^\top \tag{14}$$

The empirical estimator learned from the training examples $\{(x_i, y_i)\}_{i=1}^n$ is given by

$$\mathcal{M}_1^\top = \left( \frac{1}{n} \sum_{i=1}^n x_i x_i^\top \right)^{-1} \frac{1}{n} \sum_{i=1}^n x_i y_i^\top = w e_1^\top + \left( \frac{1}{n} \sum_{i=1}^n x_i x_i^\top \right)^{-1} \frac{1}{n} \sum_{i=1}^n x_i z_i^\top \tag{15}$$

where $z_{i,1} = 0$ and $z_{i,j} = y_{i,j}$ for $j \in [2, \ldots, d]$. Using the matrix concentration inequality, we have, with a probability $1 - 2\delta$

$$\left| \frac{1}{n} \sum_{i=1}^n x_i x_i^\top - I \right|_2 \leq \sqrt{\frac{6d}{n} \log \frac{d}{\delta}} := \epsilon_a \tag{16}$$

Since

$$\left| \sum_{i=1}^n z_i x_i^\top \right|^2 \sim \sigma^2 \chi_{n(d-1)}^2 \tag{17}$$

we have, with a probability $1 - \delta$

$$\left| \sum_{i=1}^n z_i x_i^\top \right|_F^2 \geq n(d-1)\sigma^2 \left( 1 - \sqrt{\frac{3}{n(d-1)} \log \frac{1}{\delta}} \right) \tag{18}$$

As a result, with a probability at least $1 - 3\delta$, we have

$$\begin{aligned}
\frac{|\mathcal{M}_1 - \mathcal{M}_*|}{|\mathcal{M}_*|} &= \frac{1}{|\mathcal{M}_*|} \left| \left( \frac{1}{n} \sum_{i=1}^n x_i x_i^\top \right)^{-1} \frac{1}{n} \sum_{i=1}^n z_i x_i \right| \\
&\geq \frac{\sigma}{|w|} \left( 1 + \sqrt{\frac{6d}{n} \log \frac{d}{\delta}} \right)^{-1} \sqrt{\frac{d-1}{n} \left( 1 - \sqrt{\frac{3}{n(d-1)} \log \frac{1}{\delta}} \right)} \\
&= \Omega \left( \frac{\sigma}{|w|} \sqrt{\frac{d}{n}} \right)
\end{aligned} \tag{19}$$

Our second approach is to first learn one dimension representation for each $y_i$ by minimizing the reconstruction error, and then learn a linear regression model from the derived one dimension representation. The one dimension representation is derived by minimizing the following reconstruction error:

$$\min_{u \in \mathbb{R}^n, v \in \mathbb{R}^d} |Y - vu^\top|_F^2 \tag{20}$$

where $Y = (y_1, \ldots, y_n) \in \mathbb{R}^{d \times n}$. Evidently, $v$ and $u$ are the left and right singular vectors of $Y$ with the largest singular value. We first write $YY\top$ as

$$\frac{1}{n}YY^\top = \left(\frac{1}{n}\sum_{i=1}^{n}\left(w^\top x_i\right)^2\right)e_1e_1^\top + \frac{1}{n}\sum_{i=1}^{n}z_iz_i^\top \tag{21}$$

Using the Davis–Kahan $\sin\theta$ theorem, we have

$$\sin\theta(v, e_1) \leq \frac{\left|\frac{1}{n}\sum_{i=1}z_iz_i^\top\right|_2}{\frac{1}{n}\sum_{i=1}^{n}\left(w^\top x_i\right)^2} \tag{22}$$

Using the matrix concentration inequality, we have, with a probability $1 - 2\delta$,

$$\left|\frac{1}{n}\sum_{i=1}^{n}z_iz_i^\top\right|_2 \leq \sigma^2\left(1 + \sqrt{\frac{6d}{n}\log\frac{1}{\delta}}\right) \tag{23}$$

Since

$$\sum_{i=1}^{n}(w^\top x_i)^2 \sim |w|^2\chi_n^2 \tag{24}$$

we have, with a probability $1 - \delta$

$$\frac{1}{n}\sum_{i=1}^{n}(w^\top x_i)^2 \geq \left(1 - \sqrt{\frac{3}{n}\log\frac{1}{\delta}}\right)|w|^2 \tag{25}$$

As a result, with a probability at least $1 - 2\delta$,

$$\sin\theta(v, e_1) \leq \frac{\sigma^2}{|w|^2}\frac{1 + \sqrt{\frac{6d}{n}\log\frac{1}{\delta}}}{1 - \sqrt{\frac{3}{n}\log\frac{1}{\delta}}} \tag{26}$$

As a result, the one dimension representation $u = (u_1, \ldots, u_n) = Y^\top v$, where each $u_i$ is given by

$$u_i = v^\top y_i = w^\top x_i + (v - e_1)^\top y_i \tag{27}$$

Using the one dimension representation $u$, we obtain a new estimator $\mathcal{M}_2$ given by

$$\mathcal{M}_2^\top = \left(\frac{1}{n}\sum_{i=1}^{n}x_ix_i^\top\right)^{-1}\frac{1}{n}\sum_{i=1}^{n}x_iu_iv^\top = (v^\top e_1)wv^\top + \left(\frac{1}{n}\sum_{i=1}^{n}x_ix_i^\top\right)^{-1}\frac{1}{n}\sum_{i=1}^{n}x_iz_i^\top vv^\top \tag{28}$$

First, we have, with a probability $1 - \delta$

$$\left|(v^\top e_1)wv^\top - we_1^\top\right|_F \leq \left|1 - v^\top e_1\right||w| + |w||v - e_1|$$

$$\leq \left(|1 - \cos\theta(v, e_1)| + \sqrt{\sin^2\theta(v, e_1) + (1 - \cos\theta(v, e_1))^2}\right)|w|$$

$$= \left(|1 - \cos\theta(v, e_1)| + \sqrt{2 - 2\cos\theta(v, e_1)}\right)|w| = 2\left(\sin^2\frac{\theta(v, e_1)}{2} + \sin\frac{\theta(v, e_1)}{2}\right)|w| \tag{29}$$

$$\leq 4\sin\theta(v, e_1)|w| \leq \frac{4\sigma^2}{|w|}\frac{1 + \sqrt{\frac{6d}{n}\log\frac{1}{\delta}}}{1 - \sqrt{\frac{3}{n}\log\frac{1}{\delta}}}$$

Second, since

$$\left|\frac{1}{n}\sum_{i=1}^{n}x_iz_i^\top vv^\top\right|_F \leq \left|\frac{1}{n}\sum_{i=1}^{n}x_iz_i^\top\right|_2 \tag{30}$$

using the matrix concentration inequality, we have, with a probability $1 - 2\delta$,

$$\left|\frac{1}{n}\sum_{i=1}^{n}z_ix_i^\top\right|_2 \leq \frac{4\sigma}{3n}\log\frac{d}{\delta}\sqrt{\log\frac{n}{\delta}} + \sigma\sqrt{\frac{2}{n}\log\frac{d}{\delta}} \tag{31}$$

Hence, with a probability $1 - 3\delta$, we have

$$\left| \left( \frac{1}{n} \sum_{i=1}^{n} x_i x i^\top \right)^{-1} \frac{1}{n} \sum_{i=1}^{n} x_i z_i^\top v v^\top \right|_F \leq \sigma \left( 1 - \sqrt{\frac{6d}{n} \log \frac{d}{\delta}} \right)^{-1} \left( \frac{4}{3n} \log^{3/2} \frac{d}{\delta} + \sqrt{\frac{2}{n} \log \frac{d}{\delta}} \right) \tag{32}$$

Finally, with a probability $1 - 3\delta$, we have

$$\frac{|\mathcal{M}_2 - \mathcal{M}_*|_F}{|\mathcal{M}_*|_F} \leq \frac{\sigma}{|w|} \left\{ \frac{\sigma}{|w|} \frac{1 + \sqrt{\frac{6d}{n} \log \frac{1}{\delta}}}{1 - \sqrt{\frac{3}{n} \log \frac{1}{\delta}}} + \left( 1 - \sqrt{\frac{6d}{n} \log \frac{d}{\delta}} \right)^{-1} \left( \frac{4}{3n} \log^{3/2} \frac{d}{\delta} + \sqrt{\frac{2}{n} \log \frac{d}{\delta}} \right) \right\}$$

$$= \Omega \left( \frac{\sigma^2}{|w|^2} + \frac{\sigma}{|w|\sqrt{n}} \right) \tag{33}$$

where $| \cdot |_F$ denotes the Frobenius norm. And when

$$\sigma \ll |w| \sqrt{\frac{d}{n}} \tag{34}$$

we have

$$\frac{|\mathcal{M}_2 - \mathcal{M}_*|_F}{|\mathcal{M}_*|_F} \ll \frac{|\mathcal{M}_1 - \mathcal{M}_*|_F}{|\mathcal{M}_*|_F} \tag{35}$$

Therefore, under the stated assumptions, the reconstruction-based estimator $\mathcal{M}_2$ achieves a strictly tighter approximation to $\mathcal{M}^*$ than $\mathcal{M}_1$

## C   DETAILED DERIVATION OF MUTUAL INFORMATION MAXIMIZATION

This section provides a detailed derivation showing that our proposed method, TimeAlign, is grounded in the principle of Mutual Information Maximization (MIM). We demonstrate that our loss functions, $\mathcal{L}_{\text{pred}}$ and $\mathcal{L}_{\text{align}}$, are formulated to maximize the mutual information between the representation $\mathbf{H}_x$ and the ground truth $\mathbf{Y}$.

### C.1   DERIVATION OF PREDICTION LOSS $\mathcal{L}_{\text{PRED}}$

Here, we illustrate the entire derivation of the mutual information $I(\mathbf{Y}; \mathbf{H}_x)$ in Eq. 11. We begin by deriving the prediction loss, $\mathcal{L}_{\text{pred}}$. As presented in Eq. 11, this loss is directly related to the mutual information term $I(\mathbf{Y}; \mathbf{H}_x)$. Using variational inference, we establish a lower bound for this term:

$$\begin{aligned} I(\mathbf{Y}; \mathbf{H}_x) &= \mathbb{E}_{p(y, h_x)} \left[ \log \frac{p(y, h_x)}{p(y) \cdot p(h_x)} \right], \\ &= \mathbb{E}_{p(y, h_x)} \left[ \log \frac{p(y|h_x) \cdot p(h_x)}{p(y) \cdot p(h_x)} \right], \\ &= \mathbb{E}_{p(y, h_x)} \left[ \log \frac{p(y|h_x)}{p(y)} \right], \\ &= \mathbb{E}_{p(y, h_x)} \left[ \log \frac{p(y|h_x) \cdot \mathcal{F}_\theta(y|h_x)}{p(y) \cdot \mathcal{F}_\theta(y|h_x)} \right], \\ &= \mathbb{E}_{p(y, h_x)} \left[ \log \frac{\mathcal{F}_\theta(y|h_x)}{p(y)} \right] + \mathbb{E}_{p(y, h_x)} \left[ \log \frac{p(y|h_x)}{\mathcal{F}_\theta(y|h_x)} \right]. \end{aligned} \tag{36}$$

The second term is the Kullback-Leibler (KL) divergence between the true posterior $p(y|h_x)$ and the variational approximation $\mathcal{F}_\theta(y|h_x)$. The equation can thus be rewritten as follows:

$$\begin{aligned} I(\mathbf{Y}; \mathbf{H}_x) &= \mathbb{E}_{p(y, h_x)} \left[ \log \frac{\mathcal{F}_\theta(y|h_x)}{p(y)} \right] + \int_{h_x} \int_y p(y, h_x) \cdot \log \frac{p(y|h_x)}{\mathcal{F}_\theta(y|h_x)} \, dy \, dh_x, \\ &= \mathbb{E}_{p(y, h_x)} \left[ \log \frac{\mathcal{F}_\theta(y|h_x)}{p(y)} \right] + \int_{h_x} \int_y p(y|h_x) \cdot p(h_x) \cdot \log \frac{p(y|h_x)}{\mathcal{F}_\theta(y|h_x)} \, dy \, dh_x, \\ &= \mathbb{E}_{p(y, h_x)} \left[ \log \frac{\mathcal{F}_\theta(y|h_x)}{p(y)} \right] + \int_{h_x} p(h_x) \cdot D_{\text{KL}}[p(y|h_x) || \mathcal{F}_\theta(y|h_x)] \, dh_x. \end{aligned} \tag{37}$$

Since the KL divergence term is non-negative ($D_{KL}[\cdot||\cdot] \geq 0$), we can establish a lower bound for the mutual information as follows:

$$
\begin{aligned}
I(\mathbf{Y}; \mathbf{H}_x) &= \mathbb{E}_{p(y,h_x)} \left[ \log \frac{\mathcal{F}_\theta(y|h_x)}{p(y)} \right] + \int_{h_x} p(h_x) \cdot D_{\text{KL}}[p(y|h_x)||\mathcal{F}_\theta(y|h_x)] \, dh_x, \\
&\geq \mathbb{E}_{p(y,h_x)} \left[ \log \frac{\mathcal{F}_\theta(y|h_x)}{p(y)} \right], \\
&= \mathbb{E}_{p(y,h_x)} \left[ \log \mathcal{F}_\theta(y|h_x) \right] - \mathbb{E}_{p(y,h_x)} \left[ \log p(y) \right], \\
&\geq \mathbb{E}_{p(y,h_x)} \left[ \log \mathcal{F}_\theta(y|h_x) \right] \overset{\text{def}}{=} -\mathcal{L}_{\text{pred}}.
\end{aligned}
\tag{38}
$$

As we assume that prediction errors follow a Laplace distribution with fixed scale parameter $b$:

$$
y = \hat{y} + \epsilon, \quad \text{where } \epsilon \sim \left( \frac{1}{2b} \right)^T \exp \left( -\frac{\|\epsilon - 0\|_1}{b} \right).
\tag{39}
$$

Therefore, $\mathcal{F}_\theta(y|h_x) = \left( \frac{1}{2b} \right)^T \exp \left( -\frac{\|y-\hat{y}\|_1}{b} \right)$. The derived prediction loss can be further simplified to the form of MAE loss:

$$
\mathcal{L}_{\text{pred}} = -\mathbb{E}_{p(y,h_x)} \left[ -T \log(2b) - \frac{1}{b} \|y - \hat{y}\|_1 \right] \propto \mathbb{E}_{p(y,h_x)} \left[ \|y - \hat{y}\|_1 \right],
\tag{40}
$$

where $\hat{y}$ denotes the predict results generated by the model, and $T$ is the forecasting horizon.

## C.2 DERIVATION OF ALIGNMENT LOSS $\mathcal{L}_{\text{ALIGN}}$

In addition to the prediction loss, we introduce an alignment loss, $\mathcal{L}_{\text{align}}$, to further maximize the mutual information between the representation $\mathbf{H}_x$ and the ground truth $\mathbf{Y}$. Inspired by contrastive learning, we derive an alternative lower bound for $I(\mathbf{Y}; \mathbf{H}_x)$ based on the InfoNCE objective (Oord et al., 2018):

$$
\begin{aligned}
I(\mathbf{Y}; \mathbf{H}_x) &= \mathbb{E}_{p(y,h_x)} \left[ \log \left( \frac{p(y, h_x)}{p(y) \cdot p(h_x)} \right) \right], \\
&= \mathbb{E}_{p(y,h_x)} \left[ \log \left( \frac{p(y|h_x) \cdot p(h_x)}{p(y) \cdot p(h_x)} \right) \right], \\
&= \mathbb{E}_{p(y,h_x)} \left[ \log \left( \frac{p(y|h_x)}{p(y)} \right) \right], \\
&= -\mathbb{E}_{p(y,h_x)} \left[ \log \left( \frac{p(y)}{p(y|h_x)} \right) \right], \\
&= -\mathbb{E}_{p(y,h_x)} \left[ \log \left( \frac{p(y)}{p(y|h_x)} \cdot N \right) - \log N \right], \\
&\approx -\mathbb{E}_{p(y,h_x)} \left[ \log \left( \frac{p(y)}{p(y|h_x)} \cdot N \right) \right], \\
&\geq -\mathbb{E}_{p(y,h_x)} \left[ \log \left( 1 + \frac{p(y)}{p(y|h_x)} \cdot (N-1) \cdot 1 \right) \right], \\
&= -\mathbb{E}_{p(y,h_x)} \left[ \log \left( 1 + \frac{p(y)}{p(y|h_x)} \cdot (N-1) \cdot \mathbb{E}_{p(y_j)} \left( \frac{p(y_j|h_x)}{p(y_j)} \right) \right) \right], \\
&= \mathbb{E}_{p(y,h_x)} \left[ \log \left( \frac{\frac{p(y|h_x)}{p(y)}}{\frac{p(y|h_x)}{p(y)} + \sum_{y_j \in \mathbf{Y}^{\text{neg}}} \frac{p(y_j|h_x)}{p(y_j)}} \right) \right], \\
&= \mathbb{E}_{p(y,h_x)} \left[ \log \left( \frac{f(y, h_x)}{f(y, h_x) + \sum_{y_j \in \mathbf{Y}^{\text{neg}}} f(y_j, h_x)} \right) \right].
\end{aligned}
\tag{41}
$$

Here, $f(y, h_x) = \exp(\text{proj}(h_x)^\top \cdot \mathcal{G}_\phi(y))$ is a density ratio that is proportional to $\frac{p(y|h_x)}{p(y)}$. Following common practice (Choi & Lee, 2023), we use an in-batch negative sampling strategy, where negative samples are drawn from other instances in the same mini-batch.

Furthermore, inspired by recent advances in contrastive learning (He et al., 2020; Chen & He, 2021), we simplify Eq. 41 as follows:

$$
\begin{aligned}
I(\mathbf{Y}; \mathbf{H}_x) &\approx -\mathbb{E}_{p(y,h_x)}\left[f(y, h_x)\right], \\
&= -\mathbb{E}_{p(y,h_x)}\left[\exp\left(\text{proj}(h_x)^\top \cdot \mathcal{G}_\phi(y)\right)\right] \overset{\text{def}}{=} -\mathcal{L}_{\text{align}}.
\end{aligned}
\tag{42}
$$

Therefore, we get a simpler alignment loss in Eq. 12, which further enhances the mutual information between the representation $\mathbf{H}_x$ and the ground truth $\mathbf{Y}$.

## D  EXPERIMENT DETAILS

### D.1  METRIC DETAILS

We use Mean Squared Error (MSE) and Mean Absolute Error (MAE) as evaluation metrics. Given the ground truth values $\mathbf{Y}_i$ and the predicted values $\hat{\mathbf{Y}}_i$, these metrics are defined as follows:

$$
\text{MSE} = \frac{1}{N}\sum_{i=1}^{N}(\mathbf{Y}_i - \hat{\mathbf{Y}}_i)^2, \quad \text{MAE} = \frac{1}{N}\sum_{i=1}^{N}|\mathbf{Y}_i - \hat{\mathbf{Y}}_i|
\tag{43}
$$

where $N$ is the total number of predictions.

### D.2  DATASETS

Time series forecasting is a fundamental problem with a wide range of applications in areas such as climate modeling (Niu et al., 2025; Liu et al., 2024a), finance analysis (Hu et al., 2025a;b), traffic flow prediction (Shu et al., 2021; Miao et al., 2024; Qiu et al., 2025) and so on. From these areas, we choose several widely-used time series datasets and conduct extensive experiments on TSF. We report the statistics in Tab. 4. Detailed descriptions of these datasets are as follows:

(1) **ETT** (Electricity Transformer Temperature) dataset (Zhou et al., 2021) encompasses temperature and power load data from electricity transformers in two regions of China, spanning from 2016 to 2018. This dataset has two granularity levels: ETTh (hourly) and ETTm (15 minutes).

(2) **Weather** dataset (Wu et al., 2023) captures 21 distinct meteorological indicators in Germany, meticulously recorded at 10-minute intervals throughout 2020. Key indicators in this dataset include air temperature, visibility, among others, offering a comprehensive view of the weather dynamics.

(3) **Electricity** dataset (Wu et al., 2023) features hourly electricity consumption records in kilowatt-hours (kWh) for 321 clients. Sourced from the UCL Machine Learning Repository, this dataset covers the period from 2012 to 2014, providing valuable insights into consumer electricity usage patterns.

(4) **Traffic** dataset (Wu et al., 2023) includes data on hourly road occupancy rates, gathered by 862 detectors across the freeways of the San Francisco Bay area. This dataset, covering the years 2015 to 2016, offers a detailed snapshot of traffic flow and congestion.

(5) **Solar-Energy** dataset (Liu et al., 2024b) contains solar power production data recorded every 10 minutes throughout 2006 from 137 photovoltaic (PV) plants in Alabama.

### D.3  IMPLEMENTATION DETAILS

All experiments are implemented in PyTorch (Paszke et al., 2019) and conducted on one NVIDIA V100 32GB GPU. We use the Adam optimizer (Kingma, 2014) with a learning rate selected from $\{1e^{-3}, 5e^{-4}, 1e^{-4}\}$. The batch size is set to 16 for the Electricity and Traffic datasets, and 32 for all other datasets. Tab. 5 provides detailed hyperparameter settings for each dataset.

| Dataset | Dim | Prediction Length | Dataset Size | Frequency |
|---|---|---|---|---|
| ETTm1 | 7 | $\{96, 192, 336, 720\}$ | $(34465, 11521, 11521)$ | 15 min |
| ETTm2 | 7 | $\{96, 192, 336, 720\}$ | $(34465, 11521, 11521)$ | 15 min |
| ETTh1 | 7 | $\{96, 192, 336, 720\}$ | $(8545, 2881, 2881)$ | 1 hour |
| ETTh2 | 7 | $\{96, 192, 336, 720\}$ | $(8545, 2881, 2881)$ | 1 hour |
| Electricity | 321 | $\{96, 192, 336, 720\}$ | $(18317, 2633, 5261)$ | 1 hour |
| Traffic | 862 | $\{96, 192, 336, 720\}$ | $(12185, 1757, 3509)$ | 1 hour |
| Weather | 21 | $\{96, 192, 336, 720\}$ | $(36792, 5271, 10540)$ | 10 min |
| Solar-Energy | 137 | $\{96, 192, 336, 720\}$ | $(36601, 5161, 10417)$ | 10 min |

Table 4: Dataset detailed descriptions. "Dataset Size" denotes the total number of time points in (Train, Validation, Test) split respectively. "Prediction Length" denotes the future time points to be predicted. "Frequency" denotes the sampling interval of time points.

| Tasks | Dataset | e_layers | lr | d_model | d_ff | Num. of Patches | Num. of Epochs |
|---|---|---|---|---|---|---|---|
| | ETTm1 | 2 | $1e$-4 | 128 | 256 | 1 | 10 |
| | ETTm2 | 2 | $1e$-4 | 128 | 128 | 12 | 10 |
| | ETTh1 | 2 | $5e$-4 | 32 | 32 | 24 | 10 |
| Long-term | ETTh2 | 2 | $5e$-4 | 32 | 32 | 48 | 10 |
| Forecasting | Weather | 2 | $1e$-4 | 128 | 256 | 48 | 10 |
| | Electricity | 2 | $5e$-4 | 512 | 2048 | 1 | 10 |
| | Traffic | 2 | $1e$-3 | 512 | 2048 | 1 | 30 |
| | Solar-Energy | 2 | $5e$-4 | 256 | 256 | 1 | 10 |

Table 5: Hyperparameter settings for different datasets. "e_layers" denotes the number of graph block. "lr" denotes the learning rate. "d_model" and "d_ff" denote the model dimension of attention layers and feed-forward layers, respectively.

# E  FULL RESULTS

## E.1  MAIN EXPERIMENTS

Tab. 6 and Fig. 6 present the full results for long-term forecasting, where the results are obtained through hyperparameter search. The look-back horizons is searched from $\{192, 336, 512, 720\}$. TimeAlign consistently achieves the best performance, demonstrating its effectiveness and robustness.

## E.2  ERROR BARS

In this paper, we repeat all the experiments three times. Here we report the standard deviation of our proposed TimeAlign and the second best model TVNet (Li et al., 2025), as well as the statistical significance test in Tab. 7. We perform the Wilcoxon test with TVNet (Li et al., 2025) and obtain the p-value of $1.37e^{-8}$, indicating a significant improvement at the 99% confidence level.

## E.3  ABLATION STUDIES

Tab. 8 presents the full results of the ablation studies discussed in the main text. ① **w/o Align** signifies the

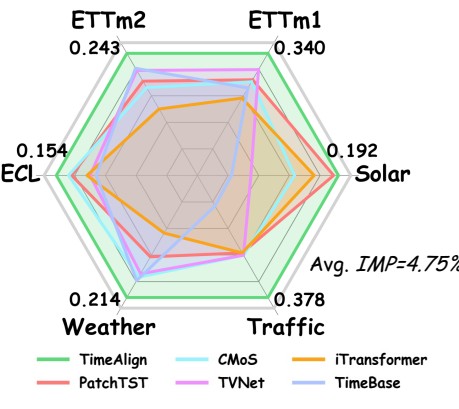

Figure 6: MSE Performance of TimeAlign.

Predict Branch alone, with both Local and Global
Alignment removed. ② **w/ Local only** denotes the Local Alignment module is retained to refine point-to-point similarity, while Global Alignment is disabled. ③ **w/ Global only** denotes only the Global Alignment is retained to coarsely align the embedding distributions, while Local Alignment is omitted. ④ **w/ Align** encompasses both Local and Global Alignment mechanisms.

### E.4 PLUG-AND-PLAY EXPERIMENT

Tab. 9 presents the full results for Plug-and-Play experiment. TimeAlign enhances forecasting accuracy across diverse benchmarks while improving alignment between predicted and ground-truth sequences in the representation space. This combined improvement in predictive precision and distributional consistency results in more reliable forecasts.

### E.5 EXTRA ANALYSIS OF RECONSTRUCTION

To validate the effect of reconstruction, we visualize the patch-wise similarity of history, ground truth, reconstructed series, and forecasts with and without alignment on the ETTm1, ECL, and

| Models | | TimeAlign (Ours) | | CMoS (2025) | | TimeBase (2025) | | TVNet (2025) | | iTransformer (2024b) | | TimeMixer (2024) | | Leddam (2024) | | ModernTCN (2024) | | PatchTST (2023) | | Crossformer (2023) | | TimesNet (2023) | | DLinear (2023) | |
|---|---|---|---|---|---|---|---|---|---|---|---|---|---|---|---|---|---|---|---|---|---|---|---|---|---|
| | Metric | MSE | MAE | MSE | MAE | MSE | MAE | MSE | MAE | MSE | MAE | MSE | MAE | MSE | MAE | MSE | MAE | MSE | MAE | MSE | MAE | MSE | MAE | MSE | MAE |
| ETTm1 | 96 | **0.279** | **0.330** | 0.292 | 0.345 | 0.311 | 0.351 | _0.288_ | _0.343_ | 0.300 | 0.353 | 0.293 | 0.345 | 0.294 | 0.347 | 0.292 | 0.346 | 0.293 | 0.346 | 0.310 | 0.361 | 0.338 | 0.375 | 0.300 | 0.345 |
| | 192 | **0.322** | **0.356** | 0.334 | _0.366_ | 0.338 | 0.371 | _0.326_ | 0.367 | 0.345 | 0.382 | 0.335 | 0.372 | 0.334 | 0.370 | 0.332 | 0.368 | 0.333 | 0.370 | 0.363 | 0.402 | 0.371 | 0.387 | 0.336 | _0.366_ |
| | 336 | **0.350** | **0.375** | 0.366 | _0.386_ | _0.364_ | _0.386_ | 0.365 | 0.391 | 0.374 | 0.398 | 0.368 | _0.386_ | 0.392 | 0.425 | 0.365 | 0.391 | 0.369 | 0.369 | 0.389 | 0.430 | 0.410 | 0.411 | 0.367 | _0.386_ |
| | 720 | **0.407** | **0.406** | 0.425 | 0.417 | 0.413 | 0.414 | _0.412_ | _0.413_ | 0.429 | 0.430 | 0.426 | 0.417 | 0.421 | 0.419 | 0.416 | 0.417 | 0.416 | 0.420 | 0.600 | 0.547 | 0.478 | 0.450 | 0.419 | 0.416 |
| | _Avg._ | **0.340** | **0.367** | 0.354 | _0.378_ | 0.357 | 0.380 | _0.348_ | 0.379 | 0.362 | 0.391 | 0.355 | 0.380 | 0.354 | 0.381 | 0.351 | 0.381 | 0.353 | 0.382 | 0.420 | 0.435 | 0.400 | 0.406 | 0.356 | _0.378_ |
| ETTm2 | 96 | **0.155** | **0.241** | 0.167 | 0.257 | 0.162 | 0.256 | _0.161_ | _0.254_ | 0.175 | 0.266 | 0.165 | 0.256 | 0.174 | 0.260 | 0.166 | 0.256 | 0.166 | 0.256 | 0.263 | 0.359 | 0.189 | 0.265 | 0.164 | 0.255 |
| | 192 | **0.210** | **0.280** | 0.228 | 0.299 | _0.218_ | _0.293_ | 0.220 | _0.293_ | 0.242 | 0.312 | 0.225 | 0.298 | 0.231 | 0.301 | 0.222 | _0.293_ | 0.223 | 0.296 | 0.345 | 0.400 | 0.254 | 0.310 | 0.224 | 0.304 |
| | 336 | **0.263** | **0.315** | 0.273 | 0.325 | _0.270_ | 0.328 | 0.272 | _0.316_ | 0.282 | 0.340 | 0.277 | 0.332 | 0.288 | 0.336 | 0.272 | 0.324 | 0.274 | 0.329 | 0.469 | 0.496 | 0.313 | 0.345 | 0.277 | 0.337 |
| | 720 | **0.343** | **0.372** | 0.367 | 0.385 | 0.352 | 0.380 | _0.349_ | _0.379_ | 0.378 | 0.398 | 0.360 | 0.387 | 0.368 | 0.386 | 0.351 | 0.381 | 0.362 | 0.385 | 0.996 | 0.750 | 0.413 | 0.402 | 0.371 | 0.401 |
| | _Avg._ | **0.243** | **0.302** | 0.259 | 0.316 | _0.250_ | 0.314 | 0.251 | _0.311_ | 0.269 | 0.329 | 0.257 | 0.318 | 0.265 | 0.320 | 0.253 | 0.314 | 0.256 | 0.317 | 0.518 | 0.501 | 0.292 | 0.330 | 0.259 | 0.324 |
| ETTh1 | 96 | 0.362 | 0.387 | _0.361_ | **0.383** | 0.349 | _0.384_ | 0.371 | 0.408 | 0.386 | 0.405 | 0.372 | 0.401 | 0.377 | 0.394 | 0.368 | 0.394 | 0.377 | 0.397 | 0.386 | 0.426 | 0.384 | 0.402 | 0.379 | 0.403 |
| | 192 | 0.404 | 0.412 | 0.405 | **0.409** | **0.387** | _0.410_ | _0.398_ | **0.409** | 0.424 | 0.440 | 0.413 | 0.430 | 0.408 | 0.427 | 0.405 | 0.413 | 0.409 | 0.428 | 0.413 | 0.442 | 0.557 | 0.436 | 0.408 | 0.419 |
| | 336 | 0.424 | 0.430 | 0.412 | 0.420 | 0.408 | 0.418 | _0.401_ | **0.409** | 0.449 | 0.460 | 0.438 | 0.450 | 0.424 | 0.437 | **0.391** | _0.412_ | 0.431 | 0.444 | 0.440 | 0.461 | 0.491 | 0.469 | 0.440 | 0.440 |
| | 720 | **0.433** | _0.451_ | **0.433** | _0.451_ | _0.439_ | _0.446_ | 0.458 | 0.459 | 0.495 | 0.487 | 0.486 | 0.484 | 0.451 | 0.465 | 0.450 | 0.461 | 0.457 | 0.477 | 0.519 | 0.524 | 0.521 | 0.500 | 0.471 | 0.493 |
| | _Avg._ | 0.406 | 0.420 | _0.403_ | _0.416_ | **0.396** | **0.414** | 0.407 | 0.421 | 0.439 | 0.448 | 0.427 | 0.441 | 0.415 | 0.430 | 0.404 | 0.420 | 0.418 | 0.436 | 0.440 | 0.463 | 0.458 | 0.450 | 0.424 | 0.439 |
| ETTh2 | 96 | _0.266_ | **0.328** | 0.274 | 0.341 | 0.292 | 0.345 | **0.263** | _0.329_ | 0.297 | 0.348 | 0.281 | 0.351 | 0.283 | 0.345 | **0.263** | 0.332 | 0.274 | 0.337 | 0.611 | 0.557 | 0.334 | 0.370 | 0.289 | 0.353 |
| | 192 | 0.330 | _0.372_ | 0.333 | 0.383 | 0.339 | 0.387 | **0.319** | **0.372** | 0.371 | 0.403 | 0.349 | 0.387 | 0.339 | 0.381 | _0.320_ | _0.374_ | 0.348 | 0.384 | 0.703 | 0.624 | 0.404 | 0.413 | 0.383 | 0.418 |
| | 336 | 0.360 | 0.399 | 0.342 | 0.384 | 0.358 | 0.410 | **0.311** | **0.373** | 0.404 | 0.428 | 0.366 | 0.413 | 0.366 | 0.405 | _0.313_ | _0.376_ | 0.377 | 0.416 | 0.827 | 0.675 | 0.389 | 0.435 | 0.448 | 0.465 |
| | 720 | _0.388_ | _0.428_ | **0.374** | **0.423** | 0.400 | 0.448 | 0.401 | 0.434 | 0.424 | 0.444 | 0.401 | 0.436 | 0.395 | 0.436 | 0.392 | 0.433 | 0.406 | 0.441 | 1.094 | 0.775 | 0.434 | 0.448 | 0.605 | 0.551 |
| | _Avg._ | 0.336 | 0.382 | 0.331 | 0.383 | 0.345 | 0.397 | _0.324_ | **0.377** | 0.374 | 0.406 | 0.349 | 0.397 | 0.345 | 0.391 | **0.322** | _0.379_ | 0.351 | 0.404 | 0.809 | 0.658 | 0.390 | 0.427 | 0.431 | 0.447 |
| Weather | 96 | **0.140** | **0.179** | _0.144_ | _0.193_ | 0.146 | 0.198 | 0.147 | 0.198 | 0.159 | 0.208 | 0.147 | 0.198 | 0.149 | 0.200 | 0.149 | 0.200 | 0.149 | 0.198 | 0.146 | 0.212 | 0.172 | 0.220 | 0.170 | 0.230 |
| | 192 | **0.182** | **0.220** | 0.186 | _0.237_ | _0.185_ | 0.241 | 0.194 | 0.238 | 0.200 | 0.248 | 0.192 | 0.243 | 0.193 | 0.240 | 0.196 | 0.245 | 0.194 | 0.241 | 0.195 | 0.261 | 0.219 | 0.261 | 0.216 | 0.273 |
| | 336 | **0.232** | **0.262** | 0.240 | 0.281 | 0.236 | 0.281 | _0.235_ | _0.277_ | 0.253 | 0.289 | 0.247 | 0.284 | 0.241 | 0.279 | 0.238 | _0.277_ | 0.245 | 0.282 | 0.252 | 0.311 | 0.280 | 0.306 | 0.258 | 0.307 |
| | 720 | **0.307** | **0.317** | 0.311 | 0.332 | 0.309 | 0.331 | _0.308_ | 0.331 | 0.321 | 0.338 | 0.318 | _0.330_ | 0.324 | 0.338 | 0.314 | 0.334 | 0.314 | 0.334 | 0.318 | 0.363 | 0.365 | 0.359 | 0.323 | 0.362 |
| | _Avg._ | **0.214** | **0.244** | 0.220 | _0.261_ | _0.219_ | 0.263 | 0.221 | _0.261_ | 0.233 | 0.271 | 0.226 | 0.264 | 0.226 | 0.264 | 0.224 | 0.264 | 0.226 | 0.264 | 0.228 | 0.287 | 0.259 | 0.287 | 0.242 | 0.293 |
| Electricity | 96 | **0.126** | **0.216** | _0.129_ | 0.223 | 0.139 | 0.231 | 0.142 | 0.223 | 0.138 | 0.237 | 0.142 | 0.247 | 0.134 | 0.228 | _0.129_ | 0.226 | _0.129_ | _0.222_ | 0.135 | 0.237 | 0.168 | 0.272 | 0.140 | 0.237 |
| | 192 | _0.143_ | **0.233** | 0.142 | _0.236_ | 0.153 | 0.245 | 0.165 | 0.241 | 0.157 | 0.256 | 0.168 | 0.269 | 0.155 | 0.248 | _0.143_ | 0.239 | 0.147 | 0.240 | 0.160 | 0.262 | 0.184 | 0.289 | 0.152 | 0.249 |
| | 336 | **0.158** | **0.249** | _0.161_ | _0.254_ | 0.169 | 0.262 | 0.164 | 0.269 | 0.167 | 0.264 | 0.171 | 0.260 | 0.173 | 0.268 | _0.161_ | 0.259 | 0.163 | 0.259 | 0.182 | 0.282 | 0.198 | 0.300 | 0.170 | 0.267 |
| | 720 | _0.189_ | **0.278** | 0.200 | 0.288 | 0.207 | 0.294 | 0.190 | 0.284 | 0.194 | 0.286 | 0.209 | 0.313 | **0.186** | _0.282_ | 0.191 | 0.286 | 0.197 | 0.290 | 0.246 | 0.337 | 0.220 | 0.320 | 0.203 | 0.301 |
| | _Avg._ | **0.154** | **0.244** | 0.158 | _0.250_ | 0.167 | 0.0258 | 0.165 | 0.254 | 0.164 | 0.261 | 0.185 | 0.284 | 0.162 | 0.256 | _0.156_ | 0.253 | 0.159 | 0.253 | 0.181 | 0.279 | 0.192 | 0.295 | 0.166 | 0.264 |
| Traffic | 96 | **0.349** | **0.225** | 0.367 | 0.256 | 0.394 | 0.267 | 0.367 | _0.252_ | _0.363_ | 0.265 | 0.369 | 0.257 | 0.415 | 0.264 | 0.368 | 0.253 | 0.370 | 0.262 | 0.512 | 0.282 | 0.593 | 0.321 | 0.395 | 0.275 |
| | 192 | **0.365** | **0.233** | _0.379_ | _0.261_ | 0.403 | 0.271 | 0.381 | 0.262 | 0.385 | 0.273 | 0.400 | 0.272 | 0.445 | 0.277 | _0.379_ | _0.261_ | 0.386 | 0.269 | 0.501 | 0.273 | 0.617 | 0.336 | 0.407 | 0.280 |
| | 336 | **0.377** | **0.240** | 0.397 | 0.270 | 0.417 | 0.278 | _0.395_ | _0.268_ | 0.396 | 0.277 | 0.407 | 0.272 | 0.461 | 0.286 | 0.397 | 0.270 | 0.396 | 0.275 | 0.507 | 0.279 | 0.629 | 0.336 | 0.417 | 0.286 |
| | 720 | **0.422** | **0.263** | 0.442 | 0.295 | 0.456 | 0.298 | 0.442 | _0.290_ | 0.445 | 0.308 | 0.461 | 0.316 | 0.489 | 0.305 | 0.440 | 0.296 | _0.435_ | 0.295 | 0.571 | 0.301 | 0.640 | 0.350 | 0.454 | 0.308 |
| | _Avg._ | **0.378** | **0.240** | _0.396_ | 0.270 | 0.417 | 0.278 | _0.396_ | _0.268_ | 0.397 | 0.282 | 0.409 | 0.279 | 0.452 | 0.283 | _0.396_ | 0.270 | 0.397 | 0.275 | 0.523 | 0.284 | 0.620 | 0.336 | 0.418 | 0.287 |
| Solar | 96 | **0.172** | **0.200** | 0.188 | 0.241 | 0.220 | 0.269 | 0.204 | 0.260 | 0.188 | 0.232 | 0.179 | 0.232 | 0.197 | 0.241 | 0.202 | 0.263 | _0.178_ | 0.229 | 0.183 | _0.208_ | 0.219 | 0.314 | 0.197 | 0.210 |
| | 192 | **0.189** | **0.213** | 0.207 | 0.256 | 0.232 | 0.266 | 0.227 | 0.272 | _0.193_ | 0.248 | 0.201 | 0.259 | 0.231 | 0.264 | 0.223 | 0.279 | **0.189** | 0.246 | 0.208 | _0.226_ | 0.231 | 0.322 | 0.218 | 0.222 |
| | 336 | 0.204 | **0.219** | 0.219 | 0.263 | 0.247 | 0.274 | 0.241 | 0.288 | 0.203 | 0.249 | **0.190** | 0.256 | 0.216 | 0.272 | 0.241 | 0.292 | _0.198_ | 0.249 | 0.212 | _0.239_ | 0.246 | 0.337 | 0.234 | 0.231 |
| | 720 | **0.202** | **0.224** | 0.226 | 0.269 | 0.247 | 0.273 | 0.242 | 0.287 | 0.223 | 0.261 | _0.203_ | 0.261 | 0.250 | 0.281 | 0.247 | 0.292 | 0.209 | _0.256_ | 0.215 | _0.256_ | 0.280 | 0.363 | 0.243 | 0.241 |
| | _Avg._ | **0.192** | **0.214** | 0.210 | 0.257 | 0.236 | 0.270 | 0.228 | 0.277 | 0.202 | 0.248 | _0.193_ | 0.264 | 0.223 | 0.264 | 0.228 | 0.281 | 0.194 | 0.245 | 0.205 | _0.232_ | 0.244 | 0.334 | 0.224 | 0.226 |

Table 6: Full results of long-term forecasting of hyperparameter searching. All results are averaged across four different prediction lengths: $O \in \{96, 192, 336, 720\}$. The best and second-best results are highlighted in **bold** and _underlined_, respectively.

| Model | TimeAlign | | TVNet (2025) | | Confidence |
|---|---|---|---|---|---|
| Dataset | MSE | MAE | MSE | MAE | Interval |
| ETTm1 | $0.340 \pm 0.011$ | $0.367 \pm 0.008$ | $0.348 \pm 0.009$ | $0.379 \pm 0.007$ | 99% |
| ETTm2 | $0.243 \pm 0.006$ | $0.302 \pm 0.006$ | $0.250 \pm 0.004$ | $0.311 \pm 0.005$ | 99% |
| ETTh1 | $0.406 \pm 0.009$ | $0.420 \pm 0.007$ | $0.407 \pm 0.010$ | $0.421 \pm 0.009$ | 99% |
| ETTh2 | $0.336 \pm 0.014$ | $0.382 \pm 0.012$ | $0.324 \pm 0.011$ | $0.377 \pm 0.015$ | 99% |
| Weather | $0.214 \pm 0.009$ | $0.244 \pm 0.007$ | $0.221 \pm 0.008$ | $0.261 \pm 0.008$ | 99% |
| Electricity | $0.154 \pm 0.013$ | $0.244 \pm 0.008$ | $0.165 \pm 0.008$ | $0.254 \pm 0.006$ | 99% |
| Traffic | $0.378 \pm 0.010$ | $0.240 \pm 0.011$ | $0.396 \pm 0.010$ | $0.268 \pm 0.013$ | 99% |
| Solar | $0.192 \pm 0.012$ | $0.214 \pm 0.013$ | $0.228 \pm 0.011$ | $0.277 \pm 0.009$ | 99% |

Table 7: Standard deviation and statistical tests for TimeAlign and second-best method (TVNet) on ETT, Weather, Electricity, Traffic and Solar datasets.

| Local Align | Global Align | | ETTm1 | | ETTm2 | | Weather | | Electricity | | Traffic | |
|---|---|---|---|---|---|---|---|---|---|---|---|---|
| w/ | w/ | Length | MSE | MAE | MSE | MAE | MSE | MAE | MSE | MAE | MSE | MAE |
| × | × | 96 | 0.289 | 0.333 | 0.162 | 0.246 | 0.151 | 0.191 | 0.129 | 0.219 | 0.358 | 0.235 |
| | | 192 | 0.328 | 0.358 | 0.221 | 0.287 | 0.193 | 0.231 | 0.146 | 0.235 | 0.377 | 0.245 |
| | | 336 | 0.359 | 0.377 | 0.275 | 0.324 | 0.243 | 0.272 | 0.162 | 0.252 | 0.393 | 0.253 |
| | | 720 | 0.420 | 0.412 | 0.349 | 0.375 | 0.314 | 0.323 | 0.200 | 0.284 | 0.431 | 0.273 |
| | | *Avg.* | 0.349 | 0.370 | 0.252 | 0.308 | 0.225 | 0.254 | 0.159 | 0.248 | 0.390 | 0.249 |
| ✓ | × | 96 | 0.284 | 0.335 | 0.158 | 0.243 | 0.145 | 0.184 | 0.129 | 0.219 | 0.352 | 0.230 |
| | | 192 | 0.324 | 0.359 | 0.212 | 0.282 | 0.188 | 0.226 | 0.145 | 0.235 | 0.372 | 0.239 |
| | | 336 | 0.356 | 0.380 | 0.265 | 0.318 | 0.236 | 0.267 | 0.160 | 0.251 | 0.385 | 0.244 |
| | | 720 | 0.414 | 0.415 | 0.345 | 0.375 | 0.310 | 0.319 | 0.196 | 0.282 | 0.425 | 0.265 |
| | | *Avg.* | 0.344 | 0.372 | 0.245 | 0.305 | 0.220 | 0.249 | 0.157 | 0.247 | 0.383 | 0.244 |
| × | ✓ | 96 | 0.282 | 0.332 | 0.157 | 0.242 | 0.142 | 0.180 | 0.128 | 0.218 | 0.354 | 0.232 |
| | | 192 | 0.325 | 0.358 | 0.211 | 0.281 | 0.184 | 0.223 | 0.146 | 0.234 | 0.369 | 0.236 |
| | | 336 | 0.353 | 0.378 | 0.265 | 0.318 | 0.235 | 0.265 | 0.161 | 0.251 | 0.383 | 0.245 |
| | | 720 | 0.409 | 0.410 | 0.343 | 0.371 | 0.311 | 0.320 | 0.193 | 0.280 | 0.424 | 0.265 |
| | | *Avg.* | 0.342 | 0.369 | 0.244 | 0.303 | 0.218 | 0.247 | 0.157 | 0.246 | 0.383 | 0.245 |
| ✓ | ✓ | 96 | 0.279 | 0.330 | 0.155 | 0.241 | 0.140 | 0.179 | 0.126 | 0.216 | 0.349 | 0.225 |
| | | 192 | 0.322 | 0.356 | 0.210 | 0.280 | 0.182 | 0.220 | 0.143 | 0.233 | 0.365 | 0.233 |
| | | 336 | 0.350 | 0.375 | 0.263 | 0.315 | 0.232 | 0.262 | 0.158 | 0.249 | 0.377 | 0.240 |
| | | 720 | 0.407 | 0.406 | 0.343 | 0.372 | 0.307 | 0.317 | 0.189 | 0.278 | 0.422 | 0.263 |
| | | *Avg.* | **0.340** | **0.367** | **0.243** | **0.302** | **0.214** | **0.244** | **0.154** | **0.244** | **0.378** | **0.240** |

Table 8: Full results of ablation on the effect of removing alignment in local and global perspective. ✓ indicates the use of alignment, while × means alignment is retained.

Traffic datasets as in Fig. 8. The similarity maps of the ground truth and the reconstructed series are almost identical, with reconstruction errors approaching zero. This demonstrates that reconstruction effectively captures multi-frequency information, including both low-frequency periodic patterns and high-frequency variations, thereby assisting the forecaster in making more accurate predictions.

## E.6 ROBUSTNESS ANALYSIS

For robustness against abrupt fluctuations, we also include experiments on high-volatility Stock Market data. It is well known that stock market data is extremely non-stationary, exhibiting evolving dynamics, abrupt regime changes, and distributional instability over time. To further evaluate robustness under distribution shifts, we conducted additional experiments on the two stock datasets (NASDAQ and NYSE) released by TFB (Qiu et al., 2024). As shown in Tab. 10, TimeAlign consistently achieves the best forecasting performance, outperforming all baselines across all prediction horizons. These findings further validate that our distilled "future prior" behaves reliably under distribution shifts, non-stationary temporal patterns, and evolving dynamics.

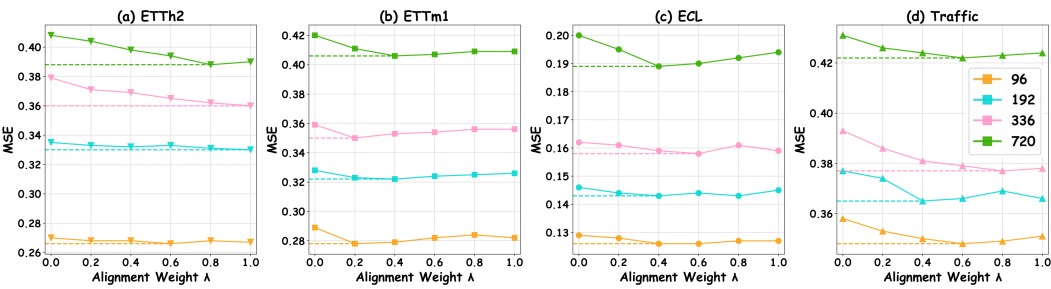

Figure 7: Effect of alignment loss weight across ETTh2, ETTm1, ECL and Traffic.

| Models | iTransformer | | | +TimeAlign | | | | DLinear | | | +TimeAlign | | | |
|---|---|---|---|---|---|---|---|---|---|---|---|---|---|---|
| Metric | MSE | MAE | SIM | MSE | MAE | SIM | $\Delta IMP$ | MSE | MAE | SIM | MSE | MAE | SIM | $\Delta IMP$ |
| **ETTm1** 96 | 0.300 | 0.353 | 0.880 | 0.302 | 0.352 | 0.883 | −0.67% | 0.300 | 0.345 | 0.887 | 0.300 | 0.342 | 0.888 | +0.00% |
| 192 | 0.345 | 0.382 | 0.870 | 0.337 | 0.374 | 0.872 | +2.32% | 0.336 | 0.366 | 0.878 | 0.332 | 0.361 | 0.880 | +1.19% |
| 336 | 0.374 | 0.398 | 0.859 | 0.364 | 0.389 | 0.863 | +2.67% | 0.367 | 0.386 | 0.863 | 0.362 | 0.380 | 0.867 | +1.36% |
| 720 | 0.429 | 0.430 | 0.840 | 0.419 | 0.421 | 0.843 | +2.33% | 0.419 | 0.416 | 0.848 | 0.413 | 0.409 | 0.853 | +1.43% |
| *Avg.* | 0.362 | 0.391 | 0.862 | 0.355 | 0.384 | 0.865 | +1.80% | 0.356 | 0.378 | 0.869 | 0.352 | 0.373 | 0.872 | +1.05% |
| **ETTm2** 96 | 0.175 | 0.266 | 0.980 | 0.170 | 0.257 | 0.984 | +2.86% | 0.164 | 0.255 | 0.982 | 0.162 | 0.248 | 0.985 | +1.22% |
| 192 | 0.242 | 0.312 | 0.974 | 0.234 | 0.298 | 0.979 | +3.31% | 0.224 | 0.304 | 0.977 | 0.218 | 0.287 | 0.979 | +2.68% |
| 336 | 0.282 | 0.340 | 0.969 | 0.278 | 0.332 | 0.972 | +1.42% | 0.277 | 0.337 | 0.971 | 0.270 | 0.325 | 0.975 | +2.53% |
| 720 | 0.378 | 0.398 | 0.956 | 0.358 | 0.381 | 0.961 | +5.29% | 0.371 | 0.401 | 0.960 | 0.359 | 0.385 | 0.963 | +3.23% |
| *Avg.* | 0.269 | 0.329 | 0.970 | 0.260 | 0.317 | 0.974 | +3.44% | 0.259 | 0.324 | 0.973 | 0.252 | 0.311 | 0.976 | +2.61% |
| **ETTh1** 96 | 0.386 | 0.405 | 0.849 | 0.382 | 0.412 | 0.853 | +1.04% | 0.379 | 0.403 | 0.862 | 0.367 | 0.394 | 0.861 | +3.17% |
| 192 | 0.424 | 0.440 | 0.833 | 0.420 | 0.439 | 0.836 | +0.94% | 0.408 | 0.419 | 0.848 | 0.406 | 0.416 | 0.846 | +0.49% |
| 336 | 0.449 | 0.460 | 0.823 | 0.447 | 0.458 | 0.825 | +0.45% | 0.440 | 0.440 | 0.838 | 0.432 | 0.440 | 0.835 | +1.82% |
| 720 | 0.495 | 0.487 | 0.795 | 0.462 | 0.482 | 0.799 | +6.67% | 0.471 | 0.493 | 0.816 | 0.452 | 0.480 | 0.819 | +4.03% |
| *Avg.* | 0.439 | 0.448 | 0.825 | 0.428 | 0.448 | 0.828 | +2.45% | 0.424 | 0.439 | 0.841 | 0.414 | 0.433 | 0.840 | +2.41% |
| **ETTh2** 96 | 0.297 | 0.348 | 0.967 | 0.288 | 0.349 | 0.971 | +3.03% | 0.289 | 0.353 | 0.970 | 0.283 | 0.344 | 0.970 | +2.08% |
| 192 | 0.371 | 0.403 | 0.959 | 0.361 | 0.395 | 0.963 | +2.70% | 0.383 | 0.418 | 0.962 | 0.362 | 0.396 | 0.962 | +5.48% |
| 336 | 0.404 | 0.428 | 0.954 | 0.394 | 0.421 | 0.959 | +2.48% | 0.448 | 0.465 | 0.956 | 0.420 | 0.439 | 0.957 | +5.36% |
| 720 | 0.424 | 0.444 | 0.950 | 0.410 | 0.442 | 0.953 | +3.30% | 0.605 | 0.551 | 0.950 | 0.607 | 0.545 | 0.951 | −0.33% |
| *Avg.* | 0.374 | 0.406 | 0.958 | 0.363 | 0.402 | 0.962 | +2.87% | 0.431 | 0.447 | 0.960 | 0.418 | 0.431 | 0.960 | +3.07% |
| **Weather** 96 | 0.159 | 0.208 | 0.852 | 0.160 | 0.203 | 0.858 | −0.63% | 0.170 | 0.230 | 0.848 | 0.169 | 0.223 | 0.845 | +0.59% |
| 192 | 0.200 | 0.248 | 0.798 | 0.209 | 0.248 | 0.809 | −4.50% | 0.216 | 0.273 | 0.805 | 0.210 | 0.256 | 0.805 | +2.78% |
| 336 | 0.253 | 0.289 | 0.751 | 0.258 | 0.294 | 0.758 | −1.98% | 0.258 | 0.307 | 0.760 | 0.255 | 0.294 | 0.758 | +1.16% |
| 720 | 0.321 | 0.338 | 0.706 | 0.334 | 0.350 | 0.682 | −4.05% | 0.323 | 0.362 | 0.698 | 0.318 | 0.346 | 0.692 | +1.55% |
| *Avg.* | 0.233 | 0.271 | 0.777 | 0.240 | 0.274 | 0.777 | −3.00% | 0.242 | 0.293 | 0.778 | 0.238 | 0.280 | 0.775 | +1.55% |
| **Electricity** 96 | 0.138 | 0.237 | 0.932 | 0.132 | 0.225 | 0.936 | +4.35% | 0.140 | 0.237 | 0.927 | 0.137 | 0.234 | 0.932 | +2.14% |
| 192 | 0.157 | 0.256 | 0.920 | 0.154 | 0.247 | 0.923 | +1.91% | 0.152 | 0.249 | 0.919 | 0.151 | 0.247 | 0.923 | +0.66% |
| 336 | 0.167 | 0.264 | 0.908 | 0.169 | 0.262 | 0.910 | −1.20% | 0.169 | 0.267 | 0.911 | 0.167 | 0.263 | 0.914 | +1.18% |
| 720 | 0.194 | 0.286 | 0.895 | 0.196 | 0.284 | 0.893 | −1.03% | 0.203 | 0.301 | 0.891 | 0.201 | 0.295 | 0.893 | +0.99% |
| *Avg.* | 0.164 | 0.261 | 0.914 | 0.163 | 0.255 | 0.916 | +0.76% | 0.166 | 0.264 | 0.912 | 0.164 | 0.260 | 0.916 | +1.20% |
| **Traffic** 96 | 0.363 | 0.265 | 0.854 | 0.354 | 0.242 | 0.857 | +2.48% | 0.395 | 0.275 | 0.838 | 0.397 | 0.275 | 0.837 | −0.51% |
| 192 | 0.385 | 0.273 | 0.848 | 0.377 | 0.255 | 0.852 | +2.08% | 0.407 | 0.280 | 0.836 | 0.406 | 0.278 | 0.835 | +0.25% |
| 336 | 0.396 | 0.277 | 0.845 | 0.399 | 0.264 | 0.846 | −0.76% | 0.417 | 0.286 | 0.834 | 0.416 | 0.283 | 0.836 | +0.24% |
| 720 | 0.445 | 0.308 | 0.831 | 0.428 | 0.279 | 0.837 | +3.82% | 0.454 | 0.308 | 0.821 | 0.452 | 0.302 | 0.822 | +0.44% |
| *Avg.* | 0.397 | 0.282 | 0.845 | 0.390 | 0.260 | 0.848 | +1.95% | 0.418 | 0.287 | 0.832 | 0.418 | 0.285 | 0.833 | +0.12% |
| **Solar** 96 | 0.188 | 0.232 | 0.860 | 0.181 | 0.230 | 0.866 | +3.72% | 0.197 | 0.210 | 0.854 | 0.190 | 0.203 | 0.858 | +3.55% |
| 192 | 0.193 | 0.248 | 0.852 | 0.205 | 0.249 | 0.851 | −6.22% | 0.218 | 0.222 | 0.849 | 0.212 | 0.217 | 0.854 | +2.75% |
| 336 | 0.203 | 0.249 | 0.850 | 0.218 | 0.250 | 0.848 | −7.39% | 0.234 | 0.231 | 0.846 | 0.224 | 0.223 | 0.851 | +4.27% |
| 720 | 0.223 | 0.261 | 0.845 | 0.210 | 0.244 | 0.847 | +5.83% | 0.243 | 0.241 | 0.844 | 0.229 | 0.233 | 0.846 | +5.76% |
| *Avg.* | 0.202 | 0.248 | 0.852 | 0.204 | 0.243 | 0.853 | −0.87% | 0.224 | 0.226 | 0.848 | 0.214 | 0.219 | 0.852 | +4.15% |

Table 9: Full results of long-term forecasting with TimeAlign as a plugin. All results are averaged across four different prediction lengths: $O \in \{96, 192, 336, 720\}$. $\Delta IMP$ denotes TimeAlign's performance gain, either *positive* or *negative*.

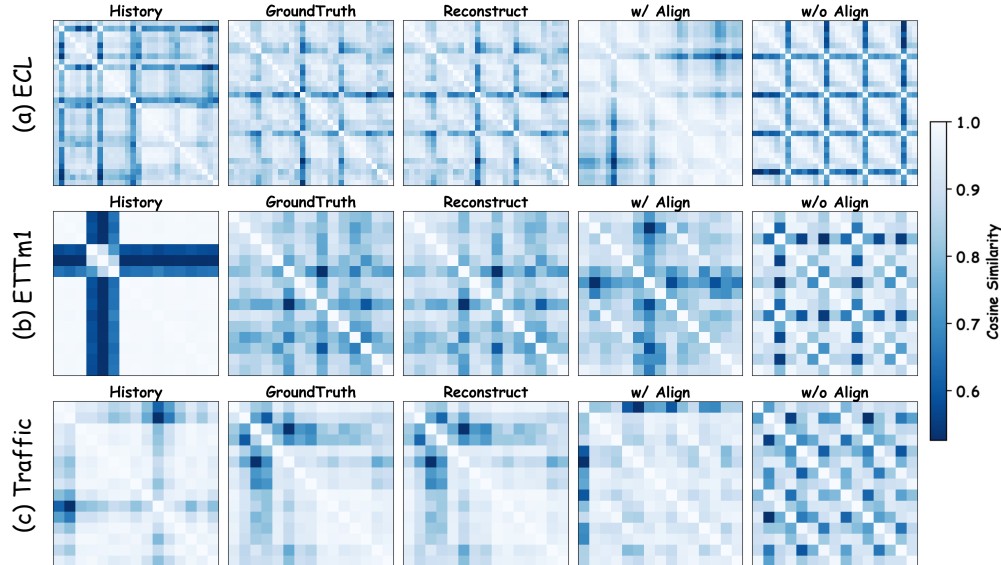

Figure 8: Visualization of patch-wise similarity of history, ground truth, reconstructed series, forecast with and without alignment from ETTm1, ECL and Traffic. The similarity maps of the ground truth and the reconstructed series are almost identical, with reconstruction errors approaching zero.

| Models | TimeAlign (Ours) | | PatchTST (2023) | | Crossformer (2023) | | DLinear (2023) | | NLinear (2023) | | MICN (2022) | | TimesNet (2023) | | FEDformer (2022) | |
|---|---|---|---|---|---|---|---|---|---|---|---|---|---|---|---|---|
| Metric | MSE | MAE | MSE | MAE | MSE | MAE | MSE | MAE | MSE | MAE | MSE | MAE | MSE | MAE | MSE | MAE |
| NASDAQ | **0.649** | **0.879** | 0.717 | 0.977 | 0.999 | 1.752 | 0.889 | 1.503 | 0.682 | 0.926 | 0.884 | 1.530 | 0.697 | 0.996 | 0.694 | 0.900 |
| NYSE | **0.385** | **0.388** | 0.427 | 0.471 | 0.913 | 0.988 | 0.526 | 0.582 | 0.401 | 0.402 | 0.639 | 0.737 | 0.475 | 0.509 | 0.422 | 0.407 |

Table 10: Long-term forecasting results on stock market data.

## E.7 HIGH FREQUENCY ANALYSIS

To directly validate TimeAlign's capability to capture rapid temporal dynamics, we introduce the High-Frequency Mean Squared Error ($\mathcal{L}_{\text{HF-MSE}}$). This metric quantifies the error between the predicted ($\hat{Y}$) and true ($Y$) series in the frequency domain, focusing only on the high-frequency band ($\mathcal{S}_{\text{high}}$) identified via knee-point detection.Calculation of HF-MSE:The metric is computed by performing the Discrete Fourier Transform (DFT) on $Y$ and $\hat{Y}$ and calculating the mean squared magnitude difference on the isolated high-frequency components:

$$\mathcal{L}_{\text{HF-MSE}} = \frac{1}{T_{\text{high}}} \sum_{k \in \mathcal{S}_{\text{high}}} \left| (Y_{\text{freq}})_k - (\hat{Y}_{\text{freq}})_k \right|^2$$

where $T_{\text{high}}$ is the count of components in $\mathcal{S}_{\text{high}}$. Tab. 11 summarizes the improvement in HF-MSE across datasets and presents results on various datasets, demonstrating the superior robustness of TimeAlign on abrupt, fine-grained temporal dynamics.

## E.8 EXTRA ANALYSIS OF DISTRIBUTION-AWARE ALIGNMENT

**Influence of Alignment Loss Weight.** We further investigate the influence of the key hyperparameter, the weight of alignment loss $\lambda$, sweeping $\lambda \in \{0.0, 0.2, 0.4, 0.6, 0.8, 1.0\}$ across four datasets. Fig. 7 shows that ETTh2 and ETTm1 are highly sensitive to $\lambda$, whereas ECL and Traffic remain relatively stable. In essence, $\lambda$ controls the trade-off between distribution matching and point-wise error minimization. As $\lambda$ increases, the model increasingly emphasizes aligning the predictive and target distributions, a strategy that generally reduces error, since better distribution overlap typically

| Models | ETTh1 | | ETTm1 | | Weather | | ECL | | Traffic | |
|---|---|---|---|---|---|---|---|---|---|---|
| Metric | HF-MSE | HF-MAE | HF-MSE | HF-MAE | HF-MSE | HF-MAE | HF-MSE | HF-MAE | HF-MSE | HF-MAE |
| TimeAlign | 2.86 | 1.27 | 2.54 | 1.19 | 3.51 | 1.35 | 40.60 | 5.09 | 253.71 | 10.64 |
| w/o Align | 2.97 | 1.36 | 2.73 | 1.35 | 3.66 | 1.50 | 42.75 | 5.44 | 261.43 | 11.56 |

Table 11: Long-term forecasting results with HF-MSE/HF-MAE metrics.

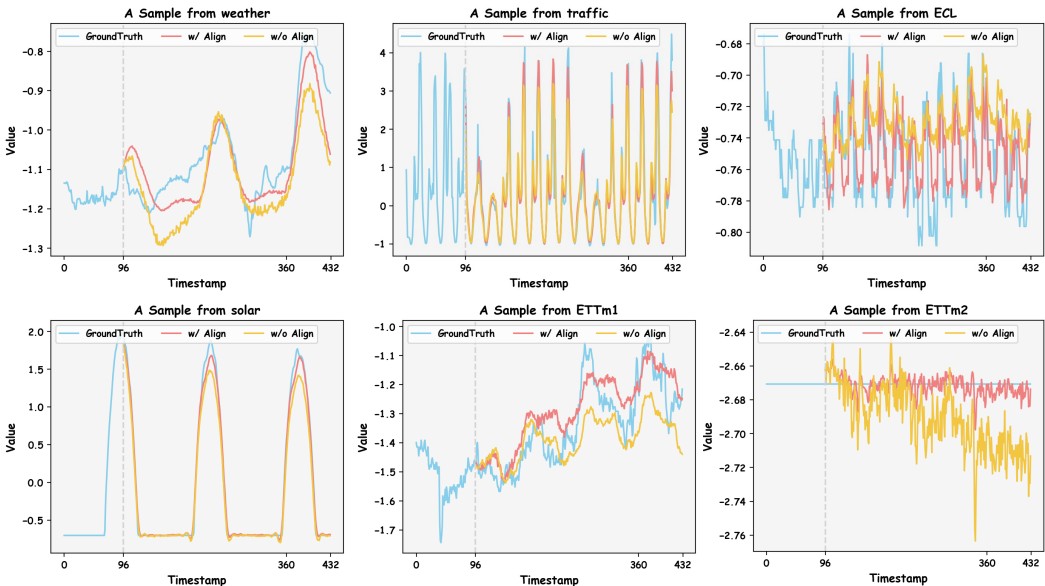

Figure 9: Visualization of predictions from ETTm1, ETTm2, ECL, Solar and Traffic.

implies lower error. However, an excessively high $\lambda$ may cause the model to highlight minority modes, particularly outliers, thereby degrading performance. Therefore, selecting an appropriate $\lambda$ for each dataset amounts to carefully balancing these two optimization objectives to maximize the overall predictive capability.

## F VISUALIZATION

In order to facilitate a clear comparison between with and without alignment, we present supplementary prediction examples for four datasets, as ETTm1, ETTm2, Weather, ECL, Solar, and Traffic in Fig. 9, respectively. The look-back horizon $L$ and the forecasting horizon $T$ are both 720 for all four datasets.

