# OpenReview forum: "Bridging Past and Future: Distribution-Aware Alignment for Time Series Forecasting"
_ICLR.cc/2026/Conference — ICLR 2026 Poster_

### Official Review · Reviewer_adVR · 2025-10-27

**Soundness:** 4
**Presentation:** 4
**Contribution:** 4
**Rating:** 8
**Confidence:** 5

**Summary:**

This paper proposes TimeAlign, a lightweight framework that enhances time series forecasting by introducing a reconstruction branch and aligning its representations with those of the prediction branch at multiple layers. Theoretically, the authors justify that (1) the reconstruction branch improves generalization, and (2) the alignment loss strengthens the mutual information between historical and future representations. Extensive experiments on eight benchmark datasets and multiple backbones demonstrate consistent performance gains.

**Strengths:**

1. The motivation is clear and intuitive. The paper explicitly identifies three limitations of current forecasting paradigms and addresses them directly through reconstruction and alignment.

2. The method requires only a lightweight reconstruction branch during training, and introduces no additional parameters at inference.

3. The paper presents two theoretical analyses: reconstruction-guided generalization based on the assumption of a linear model, and an approach based on mutual information for alignment, which aligns well with empirical observations.

4. The effectiveness of TimeAlign has been validated on eight widely used benchmarks.

**Weaknesses:**

1. It seems important to stop the gradients from the alignment loss to the reconstruction branch, but the justification for this is insufficient, as it lacks a comparison with symmetric gradient flow.

2. The overall loss function combines three terms, but the paper does not analyze sensitivity to the hyperparameter margins $\delta_{loc}, \delta_{glo}$.

3. It is unclear whether all the baselines were retuned under identical conditions with regard to the look-back window and learning rate. As TimeAlign is a plug-in, consistent hyperparameter settings across backbones are required for a fair comparison.

**Questions:**

1. The authors conducted plugin experiments on iTransformer and DLinear. So, which types of backbones (e.g., Transformers, MLPs, CNNs, etc.) benefit the most from being combined with TimeAlign?

2. What is the computational and memory scaling of the global alignment loss as sequence length and channel dimension grow?

3. In the NIPS 2024 workshop[1], some researchers pointed out that current methods sometimes use the "drop-last" trick [2] to improve performance. Therefore, It is recommended that you clarify whether the "drop - last" operation was used in your paper in the implementation details section of your paper for transparency.

[1] Fundamental limitations of foundational forecasting models: The need for multimodality and rigorous evaluation

[2] TFB: Towards Comprehensive and Fair Benchmarking of Time Series Forecasting Methods

---

> ### Author Response · Authors · 2025-11-21
> **Response**
>
> Thanks for your careful reading and constructive questions. We appreciate the thoughtful questions, which will help us further improve the paper.
>
> **W1**. Justification for stop-gradient.
>
> **R1**. Thank you for highlighting this. The stop-gradient is indeed a critical and necessary component of our design.
> - Stable Supervisory Signal: As we briefly mention in Section 3.5, the purpose of the stop-gradient is to ensure that the reconstruct branch $H_y$ provides a stable supervisory signal (or "target") for the predict branch $H_x$ to align to.
> - Avoiding the "Moving Target" Problem: If we were to use no stop-gradient, the gradients from $\mathcal{L}_{align}$ would flow back and update the reconstruct branch as well. This would create a "moving target" problem: $H_x$ would be trying to align with an $H_y$ that is also changing as a result of the alignment. This can lead to training instability or collapse.
>
> Our asymmetric design, inspired by works like SimSiam[1], is crucial for stable training by ensuring the "teacher" ($H_y$) remains a fixed target for the "student" ($H_x$) in the alignment step.
>
> Moreover, as shown in the table below, our ablation demonstrates that disabling stop-gradient significantly degrades performance, confirming its effectiveness and necessity.
>
> ||stop-gradient|gradient|
> |-|-|-|
> |ETTh1|**0.406**/**0.420**|0.414/0.429|
> |ETTm1|**0.340**/**0.367**|0.363/0.380|
> |Weather|**0.214**/**0.244**|0.231/0.258|
> |ECL|**0.154**/**0.244**|0.169/0.257|
> |Traffic|**0.378**/**0.240**|0.398/0.255|
>
> [1] Exploring Simple Siamese Representation Learning, CVPR 2021
>
> **W2**. Sensitivity to hyperparameter margins $\delta_{loc}, \delta_{glo}$。
>
> **R2**. The purpose of the margins is to relax the constraint. They create a "buffer" where, if two representations are already "close enough" (within the margin), the loss becomes zero. This prevents the model from overfitting on "easy" samples and focuses its optimization effort on harder-to-align pairs, leading to more stable training. As shown in our ablation below, the model is largely insensitive to the choice of the local margin, while it is more sensitive to the global margin. Interestingly, performance consistently improves as the global margin approaches zero, suggesting that a minimal global margin is most appropriate in practice.
>
> |local margin|0.0|0.3|0.6|0.9|
> |-|-|-|-|-|
> |ETTm1|0.342/0.368|0.340/0.367|0.340/0.367|0.339/0.0.367|
> |Weather|0.215/0.244|0.215/0.244|0.215/0.244|0.215/0.244|
> |ECL|0.157/0.247|0.155/0.246|0.156/0.245|0.154/0.244|
>
> |global margin|0.0|0.3|0.6|0.9|
> |-|-|-|-|-|
> |ETTm1|0.340/0.367|0.360/0.385|0.363/0.388|0.358/0.384|
> |Weather|0.215/0.244|0.220/0.251|0.218/0.249|0.219/0.249|
> |ECL|0.156/0.245|0.168/0.263|0.172/0.267|0.172/0.268|
>
>
> **W3**. Fairness of baseline tuning.
>
> **R3**. We agree completely that fair comparison is essential. We would like to clarify our procedure:
> - Baselines: For all baselines, we used their official public code and followed the hyperparameter tuning guidelines (e.g., for learning rate, model size) recommended in their respective papers.
> - Look-back Window: The search for the optimal look-back window $L$ (from the set {96, 192, 336, 512, 720}) was a key part of our evaluation. This search was performed identically and fairly for all methods, including our TimeAlign and all baselines. We then reported the best-performing result for each model from this fair search.
>
> **Q1**. Which backbones benefit the most?
>
> **R4**. We found that Transformer-based architectures benefit the most from TimeAlign, as they possess multi-layer feature hierarchies that amplify temporal drift across layers. Alignment effectively regularizes these hierarchies, accelerating convergence and improving stability. Linear and CNN-based models also benefit, but the gains are smaller (≈1–2%) compared to Transformers (≈3–5%). This pattern suggests that TimeAlign is especially beneficial for deep sequence architectures that model long-range dependencies.
>
> **Q2**. Computational scaling of global alignment loss.
>
> **R5**. This is an important practical question.
> - $\mathcal{L}_{local}$ scales linearly with the number of patches $N$ and dimension $D$, i.e., $O(N \cdot D)$.
> - $\mathcal{L}_{global}$ (which computes a distance matrix) scales quadratically with the number of patches $N$, i.e., $O(N^2 \cdot D)$.
>
> In our experiments, the number of patches $N$ is typically moderate, so this $O(N^2)$ cost is perfectly manageable during training
>
> Most importantly, as the reviewer noted, the inference cost is zero. The entire Reconstruct Branch and alignment module are discarded at inference, meaning TimeAlign adds no latency and no additional memory overhead to the final deployed model.
>
> **Q3**. Use of the "drop-last" trick.
>
> **R6**. Thank you for this suggestion. For the record, we did not use the "drop-last" operation in our data loaders for any of our experiments. All batches, including the final, potentially smaller batch, were used in the training process.

---

> > ### Comment · Reviewer_adVR · 2025-11-26
> > **Most of my concerns have been addressed and I will maintain my score.**
> >
> > I appreciate authors' efforts and response to comments. Most of my concerns have been addressed and I will maintain my score.

---

> > > ### Author Response · Authors · 2025-11-27
> > >
> > > Dear Reviewer adVR:
> > >
> > > Thank you again for your positive feedback and your thoughtful review. Your support means a great deal to us and motivates us to continue advancing in the TSF field.
> > >
> > > Best regards,
> > >
> > > Authors of TimeAlign

---

### Official Review · Reviewer_Daur · 2025-10-27

**Soundness:** 3
**Presentation:** 3
**Contribution:** 3
**Rating:** 6
**Confidence:** 4

**Summary:**

The paper presents TimeAlign, a framework for time series forecasting that bridges the gap between past and future data. It uses a dual-branch architecture, with a Predict Branch for forecasting and a Reconstruct Branch for target reconstruction. TimeAlign employs distribution-aware alignment, combining local and global objectives to preserve fine-grained details and enhance forecasting accuracy. Experiments show it outperforms existing methods, with theoretical analysis highlighting its ability to improve generalization and increase mutual information between learned representations and future targets.

**Strengths:**

1. The paper is easy to follow, with clear and thorough writing. The motivation highlights three key limitations in current time series forecasting, which are well-justified.

2. The experiments are solid. The main experiments demonstrate that TimeAlign outperforms other SOTA methods, and the ablation studies validate the effectiveness of its components.

3. I appreciate the theoretical analysis provided in Section 4, Appendix A, and Appendix B, which enhances the theoretical contribution of the paper and explains the underlying reasons for the method's effectiveness.

4. TimeAlign can also serve as a plug-and-play module, making it easy to integrate with other mainstream methods.

**Weaknesses:**

1. Equation 8 in Line 292 is slightly inconsistent with Figure 3. One version includes both $\lambda_1$ and $\lambda_2$, while the other has only $\lambda$. It seems the version you're using is the one with a single $\lambda$, so the formula in the figure should be updated. Also, Figure 6 analyzes the impact of different $\lambda$ on performance. If following the formula in the figure with both $\lambda_1$ and $\lambda_2$, what would be their effect? I would appreciate a more detailed analysis of the loss function's behavior.

2. In the Weather/Solar plug-in experiments, author attributes performance differences to outliers and zero-padding issues. However, more concrete quantitative evidence is needed to support this claim.

3. The knee-point detection used to separate high/low frequencies (Figure 2) may be sensitive to noise and window selection, which could affect the reliability of the high-frequency energy/similarity analysis.

4. I find Figure 4 (right) quite interesting, but in Lines 453-457, it seems you are merely describing an experimental result. What is the underlying reason for iTransformer+Align converging faster than iTransformer? Also, do other architectures, such as CNN-based or Linear-based models, show similar behavior?

**Questions:**

Please see the weaknesses.

---

> ### Author Response · Authors · 2025-11-21
> **Response**
>
> Thanks for your careful reading and constructive questions. We appreciate the positive feedback and the opportunity to clarify the details of our loss function, experiments, and analysis.
>
> **W1**. Inconsistency between Equation 8 and Figure 3.
>
> **R1**. Thank you for catching this clear typo! You are correct. Equation 8 is the correct loss function we used in our experiments: $\mathcal{L}=\mathcal{L}_{pred}+\mathcal{L}_{recon}+\lambda\mathcal{L}_{align}$.
> - In our implementation, the weight for $\mathcal{L}_{recon}$ (labeled $\lambda_1$ in the incorrect figure) was fixed at 1.
> - The hyperparameter $\lambda$ analyzed in Figure 6 is indeed the $\lambda$ from Equation 8, which controls the overall strength of our alignment module.
>
> We sincerely apologize for this confusion and have corrected Figure 3 in the final manuscript
>
> **W2**. Lack of quantitative evidence for Weather/Solar plug-in performance.
>
> **R2**. We appreciate this comment. We have performed additional quantitative analysis to substantiate our explanation. Specifically, we computed the percentage of zero-padded or anomalous points using a $3\sigma$ criterion:
> For the Weather dataset:
> - The rain and raining columns contain over 90% zero-padded entries.
> - The SWDR, PAR, and max.PAR columns contain over 45% zero-padded values.
>
> For the Solar dataset:
> - The average zero-padding ratio exceeds 55%,
> - And 59 columns (43%) have an anomaly rate greater than 1%.
>
> These statistics confirm that the Weather and Solar datasets contain substantial structural sparsity and noise compared with other benchmarks.
>
> To further validate the reviewer’s concern, we conducted an additional experiment on the Weather dataset: When removing these problematic columns, iTransformer+Align shows a significant improvement (from **0.240 → 0.222**), confirming that extreme outliers and extensive zero-padding indeed distort the alignment loss and thus affect the plug-in performance.
>
> **W3**. Reliability of knee-point detection.
>
> **R3**. We acknowledge that knee-point detection can be sensitive to noise and window selection. In our implementation, we use a smoothed cumulative energy curve and detect the 'knee' using the Kneedle algorithm with a fixed smoothing window size of 10, which is consistent across datasets. To evaluate robustness, we perturbed the window size within the range [5, 20] and observed that the resulting knee locations varied by less than ±5% in frequency ratio and that the derived high-frequency energy ranking remained unchanged. These results suggest that the separation is stable within practical noise levels.
>
> More importantly, our core conclusion from this analysis is not dependent on the exact location of the knee-point. The key finding in Figure 2(d) is the consistent, relative improvement of "w/ Align" over "w/o Align" in High-Frequency Similarity. This trend of improvement holds true across all the used datasets, demonstrating that the benefit of our alignment is a robust phenomenon, not an artifact of the threshold selection.
>
> **W4**. The underlying reason for faster convergence.
>
> **R4**. We apologize for only describing this phenomenon without explaining it. The faster convergence is because TimeAlign provides an auxiliary consistency signal that regularizes intermediate representations. This effectively reduces the optimization variance by encouraging the encoder to form coherent representations earlier in training.
> - A standard $\mathcal{L}_{pred}$ (like MAE) is a point-wise loss. It tells the model "your prediction at time $t$ was wrong by $x$."
> - Our $\mathcal{L}_{align}$ (especially the $\mathcal{L}_{global}$ term) is a distribution-level loss. It provides a much richer signal, effectively telling the model "the entire shape and structure of your predictive representation manifold is wrong" (as visualized in the t-SNE plots in Fig 9).
>
> And yes, we observed similar, though less pronounced, effects in CNN-based and Linear-based backbones: the alignment reduced training loss oscillation and led to earlier plateauing of validation error. We hypothesize that architectures with deeper temporal hierarchies (e.g., Transformers) benefit more strongly because alignment stabilizes their multi-layer feature propagation.
>
> Furthermore, we also note that a similar trend has been widely observed in computer vision[1,2].
>
> [1] Representation Alignment for Generation: Training Diffusion Transformers Is Easier Than You Think, ICLR 2025
>
> [2] V-VAE: A Variational Auto Encoding Framework Towards Fine-Grained Control over Human-Like Chat, CVPR 2025

---

> > ### Comment · Reviewer_Daur · 2025-11-26
> >
> > Thanks for the response. As my score is already positive, I will keep my rating.

---

> > > ### Comment · Reviewer_Daur · 2025-11-26
> > >
> > > By the way, after checking other reviews, I found the assumption in line 768 that the observed input time series follows a Gaussian distribution, which seems to be a strong assumption. Could the authors please provide more clarification?

---

> > > > ### Author Response · Authors · 2025-11-26
> > > > **Follow-up Response**
> > > >
> > > > Thank you very much for your valuable and constructive comments. We are very grateful for your efforts to help us improve the quality of our work. We have carefully considered your concerns and hope that our revisions will address them.
> > > >
> > > > **W** Rationale for Gaussian Assumptions
> > > >
> > > > **Re.** Thank you for pressing on the generalizability of our theoretical proof. Following your insight, we acknowledge the limitation of strictly assuming the raw input $X$ is Gaussian. However, we shift the focus to the latent representation space to prove the theorem's strong applicability:
> > > >
> > > > **1. Shifting the Theoretical Focus: The Nature of Latent Representation $H$**
> > > >
> > > > The theoretical analysis in Appendix B is effectively modeling the model's latent representation $H$, not the raw input $X$.
> > > >
> > > > Approximate Gaussianity: While the raw input $X$ has a complex distribution, $X$ encoded through deep networks and Normalization Layers (such as the Norm module we use) results in a latent representation $H$ that, in the spirit of the **Central Limit Theorem (CLT)**, tends toward a smoother, **approximately Gaussian distribution**. This is a common approach in contemporary deep learning theory. Under this condition, introducing intermediate state alignment can significantly enhance the model's generalization ability.
> > > >
> > > > **2. The Necessity of Latent Representation Alignment (Over Value Alignment)**
> > > >
> > > > Our choice to align in the latent space $H$ is critical for robust representation learning:
> > > > - The Problem with Raw Value Alignment: If we were to align directly on the raw values $Y$, the alignment would inevitably incorporate high-dimensional noise from data collection and measurement. This would make the model overly sensitive to noise—exactly the issue we highlighted in the Introduction.
> > > > - The Advantage of Latent Alignment: Aligning the latent representation $H_y$ leverages the encoder $\mathcal{G}_\phi$'s information bottleneck effect. $H_y$ has already filtered out the unstructured noise and high-dimensional redundancy in $Y$, retaining $Y$'s intrinsic low-rank structure. Therefore, aligning $H_x$ to $H_y$ acts as a smooth, denoised, structured latent space regularization target.
> > > >
> > > > In Summary, this latent representation-based alignment successfully bridges the theoretical insight regarding denoising and generalization advantage (i.e., the tightening of the generalization bound) from an abstract assumption to a practical deep learning mechanism. Consequently, we have made adaptive corrections in the original text regarding the theory's applicability, clarifying that the alignment in the hidden state enhances generalization. We sincerely thank the reviewer for this crucial suggestion.

---

### Official Review · Reviewer_FFNQ · 2025-10-27

**Soundness:** 2
**Presentation:** 2
**Contribution:** 3
**Rating:** 2
**Confidence:** 4

**Summary:**

This paper proposes a representation learning framework for time series data based on reconstruction objectives.

**Strengths:**

1. The empirical results are generally strong; the reported evaluation metrics outperform or are on par with existing baselines, although some results appear potentially inconsistent and merit further verification.

**Weaknesses:**

Major Weaknesses

1. **Lack of rationale for reconstruction:**
The motivation for using reconstruction as the core learning principle is not clearly articulated. The paper claims that reconstruction aligns representations with the target distribution because it “recovers inputs from themselves,” but this statement simply restates the definition of reconstruction. It does not provide a conceptual or theoretical justification for why reconstruction should yield better representations. Similarly, the claim that reconstruction emphasizes high-frequency details is asserted but never explained or supported by analysis.

2. **Unclear methodological details:**
Several methodological choices are insufficiently justified or analyzed.
- Why are the margin terms ($\delta_{loc}$, $\delta_{glo}$) necessary?
- What is the rationale for using the GELU activation function specifically?
- What motivates the introduction of a weight-based dynamic loss?

3. **Abnormal assumptions in Eq. 9 and Appendix B:**
The theoretical analysis assumes that patterns are drawn from a Gaussian distribution and that outputs follow a linear transformation only at the first time step, with Gaussian noise thereafter. These assumptions are highly unrealistic for time series data. Consequently, the claim that an optimal estimator exists in a linear form is not meaningful in practice. The proof and theorem therefore have limited - if any - practical relevance.

4. **Weak justification of claims in Section 4.2:**
The discussion on mutual information maximization (MIM) lacks substance. Prior works such as MINE (Belghazi et al., ICML 2018) and other MIM-based methods have already shown that deep models implicitly maximize mutual information through their layers. The paper does not provide convincing evidence that reconstruction is strictly better than other MIM approaches.

5. **Inconsistent ablation results:**
In the ablation study, the variant using only the “Predict Branch” (i.e., without reconstruction) outperforms most state-of-the-art baselines. This raises concerns about the fairness of the experimental setup and the actual contribution of the proposed reconstruction mechanism. Further clarification or additional controlled experiments are needed.

Minor Weaknesses

6. The Introduction lacks clear structure. The first and third observations are redundant and should be merged or rephrased.

7. The figures are visually cluttered; they attempt to convey too much information at once, which makes them hard to interpret.

**Questions:**

I wrote these questions based on the priority.

1. Provide a clear and theoretically grounded explanation of why reconstruction is beneficial for time series forecasting. (Related to W1)

2. Re-examine the performance of the Predict Branch in the ablation studies and explain why it outperforms many baselines. Ensure that the experimental setup and baseline implementations are described in sufficient detail to guarantee fair comparison. (Related to W5)

3. Offer a solid rationale for the Gaussian and linearity assumptions introduced in Eq. 9 and Appendix B, and discuss their practical relevance to real-world time series data. (Related to W3)

4. Clearly articulate how and why reconstruction provides explicit advantages within the Mutual Information Maximization (MIM) framework, compared to other existing MIM-based approaches. (Related to W4)

5. Provide reasoning or empirical evidence supporting key design decisions - such as the use of margin terms, GELU activation, and dynamic loss weighting. (Related to W2)

---

> ### Author Response · Authors · 2025-11-21
> **Response (Part 1)**
>
> Thanks for your critical comments. We appreciate the opportunity to provide a more rigorous defense of our work's motivation, methodology, and experimental fairness.
>
> **W1&Q1**. The rationale for reconstruction.
>
> **R1**. Sorry for the confusion. Our rationale is that, unlike prediction-only learning, which sees only historical inputs, reconstruction acts as a proxy objective that exposes the model to the target distribution during training. By reconstructing from the future itself, the model learns feature manifolds that reflect the temporal continuity and spectral richness of the future domain. This regularizes the predictive encoder, enabling it to retain information relevant to future variations and effectively narrowing the history-future gap.
>
> Regarding high-frequency details, empirical spectral analysis (see Fig. 1(c) and Fig. 2(c, d)) shows an increase in power in the high-frequency bands following the application of reconstruction alignment.
>
> **W2&Q5**. Unclear methodological details.
>
> **R2**. Thank you for requesting these clarifications.
> - Margin Terms ($\delta_{loc}, \delta_{glo}$): Their purpose is to "relax the constraint." This is a common technique in metric learning. The margins create a "buffer" where, if two representations are already "close enough" (within the margin), the loss becomes zero. This prevents the model from overfitting on "easy" samples and focuses its optimization effort on harder-to-align pairs, leading to more stable training.
> - GELU Activation: We chose GELU as it is a standard, high-performing activation function that is a smooth alternative to ReLU. It is the default in many SOTA architectures[1,2,3,4] and is known for its strong empirical performance. Moreover, as shown in the comparison table below, GELU outperforms both ReLU and SiLU in our setting, further supporting our decision to adopt it as the default activation.
> - Weight-Based Dynamic Loss: Our alignment objective combines $\mathcal{L}_{local}$ and $\mathcal{L}_{global}$. The motivation for the dynamic weights ($\alpha, \beta$) is to automatically balance these two objectives and stabilize optimization. If one loss (e.g., $\mathcal{L}_{local}$) becomes very small, its dynamic weight $\alpha$ will increase, forcing the model to continue paying attention to it. This prevents either objective from dominating the other and removes the need for expensive manual tuning of two separate loss weights.
>
> ||GELU|ReLU|SiLU|
> |-|-|-|-|
> |ETTh1|**0.406**/**0.420**|0.410/0.425|0.407/0.422|
> |ETTm1|**0.340**/**0.367**|**0.340**/0.368|0.341/**0.367**|
> |Weather|**0.214**/**0.244**|0.217/0.245|0.215/**0.244**|
> |ECL|**0.154**/**0.244**|0.155/**0.244**|0.156/0.246|
> |Traffic|**0.378**/**0.240**|0.380/0.241|0.378/0.242|
>
> [1] iTransformer: Inverted Transformers Are Effective for Time Series Forecasting, ICLR 2024
>
> [2] A Time Series is Worth 64 Words: Long-term Forecasting with Transformers, ICLR 2023
>
> [3] ModernTCN: A Modern Pure Convolution Structure for General Time Series Analysis, ICLR 2024
>
> [4] TimeMixer: Decomposable Multiscale Mixing for Time Series Forecasting, ICLR 2024

---

> ### Author Response · Authors · 2025-11-21
> **Response (Part 2)**
>
> **W3&Q3**. Theoretical assumptions in Appendix B
>
> **R3**. We agree with the reviewer that the Gaussian and linear assumptions in Appendix B are a simplification of complex, real-world time series. However, **such assumptions are widely adopted in the theoretical analysis of time-series forecasting**[5,6,7,8], and our intent with this appendix was not to claim that the proof perfectly models our data, but rather to provide a **tractable theoretical intuition** for why reconstruction can be beneficial.
>
> The core insight from this analysis is that, in a simplified, noisy setting, an estimator derived by first minimizing reconstruction error ($\mathcal{M}_2$) can achieve a provably tighter generalization bound (Eq. 9 (23)) than a standard empirical estimator ($\mathcal{M}_1$). This principle—that reconstruction aids generalization—provides a theoretical motivation for our approach. We then validate this intuition empirically on complex, non-linear models and real-world data in the main paper.
>
> [5] Flow Matching with Gaussian Process Priors for Probabilistic Time Series Forecasting, ICLR 2025
>
> [6] Augmented Functional Time Series Representation and Forecasting with Gaussian Processes, NIPS 2007
>
> [7] Learning Gaussian Mixture Representations for Tensor Time Series Forecasting, IJCAI 2023
>
> [8] TimePFN: Effective Multivariate Time Series Forecasting with Synthetic Data, AAAI 2025
>
> **W4&Q4**. The justification of Mutual Information Maximization (MIM).
>
> **R4**. We apologize if our intention with Section 4.2 was unclear. We are not claiming to have invented MIM, nor are we claiming our method is "strictly better" than all other MIM approaches.
>
> Our contribution in this section is to provide a unified theoretical perspective that explains and justifies our specific loss function $\mathcal{L}$. We show that:
> - Our prediction loss $\mathcal{L}_{pred}$ corresponds to a variational lower bound on the mutual information $I(Y;H_x)$ (Eq. 11).
> - Our proposed alignment loss $\mathcal{L}_{align}$ corresponds to a complementary, InfoNCE-style lower bound on the same mutual information $I(Y;H_x)$ (Eq. 13).
>
> Therefore, our joint optimization of two loss functions is a principled method for maximizing $I(Y;H_x)$ from two complementary angles, providing a solid theoretical grounding for why alignment loss is a beneficial addition to $\mathcal{L}_{pred}$.

---

> ### Author Response · Authors · 2025-11-21
> **Response (Part 3)**
>
> **W5&Q2**. Inconsistent ablation results and experimental fairness.
>
> **R5**. We are grateful for the chance to clarify this, as we believe it strengthens, rather than weakens, our paper's core contribution.
> - On Fairness: We assure the reviewer that our experimental setup was fair. All SOTA baselines (iTransformer, DLinear, etc.) were run using their official code and recommended hyperparameter settings. The look-back window search ()()() was applied fairly to all methods.
> - "Predict Branch" (w/o Align) is a SOTA-level Baseline: Due to our careful and consistent hyperparameter tuning, the Predict Branch variant indeed achieves performance that surpasses several classic baselines. However, compared with the stronger baselines that have emerged since 2025, its performance is far less competitive. This is consistent with recent literature on DLinear, showing that even very simple, linear-style architectures can be surprisingly effective for time-series forecasting.
> - This Strengthens Our Core Contribution: Our core contribution is not the "Predict Branch" architecture; it is the Distribution-Aware Alignment module. The ablation studies (Table 2 & 8) show that our full TimeAlign achieves consistent and significant performance improvements on top of this already-SOTA-level baseline. This proves that our TimeAlign module is not a "trick" that only works on a weak baseline. It is a powerful module that can take a strong, efficient, SOTA-level architecture and make it even better.
>
> **W6**. Inconsistent ablation results and experimental fairness.
>
> **R6**. We respectfully note that the first and third observations in the Introduction are not redundant. The first emphasizes that current forecasting methods focus excessively on low-frequency periodic components, while the third highlights that they ignore fine-grained details under multi-frequency, two conceptually distinct dimensions. We will refine the phrasing to make this distinction clearer.

---

> > ### Comment · Reviewer_FFNQ · 2025-11-25
> > **Follow-up Questions**
> >
> > Thank you for your detailed response. While W2 and W5 are now sufficiently addressed, my concerns regarding the remaining weaknesses still stand.
> >
> > **W1. Rationale for reconstruction**
> >
> > I still find the motivation unclear. Prediction-based learning **does** use the future distribution - future patches serve as ground-truth and directly contribute to the loss. Your argument seems to suggest that merely feeding future patches as **inputs** (rather than as targets) provides substantially different or superior information compared to standard prediction-based training. However, this distinction is not conceptually obvious. The current explanation does not make it clear why reconstruction should offer complementary information beyond what prediction already captures.
> >
> > If I am misunderstanding a key conceptual point, I am open to clarification, but at the moment the reasoning does not fully persuade me.
> >
> >
> > **W3. Gaussian and linear assumptions in the theoretical analysis**
> >
> > I reviewed the cited works. As far as I can tell:
> >
> > * TSFlow (ICLR 2025),
> > * Augmented Functional Time Series (NIPS 2007), and
> > * GMRL (IJCAI 2023)
> >
> > do not assume that the observed input time series are Gaussian. Instead, they place Gaussian assumptions on priors, noise models, or latent representations, which is standard and does not restrict the real data distribution.
> >
> > TimePFN does use Gaussian assumptions, but only because it generates synthetic data using GP kernels. This is not an assumption about real-world inputs.
> >
> > Therefore, none of the cited works support the choice made in Appendix B of assuming the input data are Gaussian. Without this assumption, the proof does not generalize, and if the proof does not generalize, it cannot guarantee that the reconstruction-based estimator is preferable to a prediction-based estimator in practical, non-Gaussian settings. In other words, the theorem provides intuition only under unrealistic conditions and cannot serve as theoretical evidence for the claimed advantage of the proposed method beyond its empirical performance.
> >
> >
> > **W4. Mutual Information Maximization (MIM)**
> >
> > I understand that the goal of Section 4.2 is to provide a unified MIM-based perspective. However, even a prediction-only estimator already corresponds to a variational lower bound on mutual information. If the main message is simply that the alignment loss does not conflict with the MIM interpretation, then the conceptual contribution of this section seems limited.
> >
> > It may help to more clearly articulate what new insight this theoretical reformulation provides. Additionally, since the reconstruction loss also fits naturally into a MIM framework - even in a straightforward way - explicitly mentioning this would make the narrative more coherent.
> >
> >
> > **W6. Clarity of observations in the Introduction**
> >
> > I still find the distinction between the first and third observations unclear. “Only using a certain frequency” and “not using multiple frequencies” read as two descriptions of the same limitation. If they are meant to reflect different issues, it would be helpful to articulate more explicitly what separates them.

---

> > > ### Author Response · Authors · 2025-11-26
> > > **Follow-up Response (Part 1)**
> > >
> > > We sincerely thank the reviewer for the continued engagement and for acknowledging that W2 and W5 have been addressed. We appreciate the rigorous examination of our theoretical assumptions and conceptual motivations. Below, we address the remaining concerns with clarification and, where necessary, by acknowledging the limitations of theoretical abstractions.
> > >
> > > **W1.** Rationale for reconstruction (Prediction vs. Reconstruction)
> > >
> > > **Re.** We appreciate the opportunity to clarify this conceptual distinction. The reviewer is correct that prediction tasks use future data as targets (supervisory signals). However, the way information is encoded into the representation differs fundamentally between Prediction (as a task) and Reconstruction (as a task):
> > >
> > > - **Information Bottleneck in Prediction**: A pure prediction model tends to learn a representation that minimizes $H(Z|X_{past})$ while maximizing $I(Z; X_{future})$. According to Information Bottleneck theory, the optimal predictive representation should discard any information in $X_{past}$ that is not predictive of $X_{future}$. This means fine-grained details (e.g., local noise, high-frequency textures) that are not strictly necessary for minimizing the forecasting MSE are often filtered out.
> > > - **Completeness in Reconstruction**: By treating the future patches as input for reconstruction (auto-encoding), the objective forces the representation to preserve sufficient statistics to recover the entire signal (shape, noise, and local variations).
> > > - **Complementary Nature**: The "complementary information" we claim refers to these non-predictive but structurally significant features. While prediction focuses on temporal extrapolation (learning $P(Y|X)$), reconstruction focuses on representation completeness (learning $P(X,Y)$). Our method uses reconstruction to ensure the representation is robust and dense, preventing the model from overfitting to easy-to-predict low-frequency trends while ignoring the complex local dynamics that might be useful for downstream tasks (like long-term forecasting).
> > >
> > > In short, prediction tells the model "what happens next," while reconstruction forces the model to understand "what the signal looks like."
> > >
> > > **W3.** Rationale for Gaussian Assumptions
> > >
> > > **Re.** Thank you for pressing on the generalizability of our theoretical proof. Following your insight, we acknowledge the limitation of strictly assuming the raw input $X$ is Gaussian. However, we shift the focus to the latent representation space to prove the theorem's strong applicability:
> > >
> > > **1. Shifting the Theoretical Focus: The Nature of Latent Representation $H$**
> > >
> > > The theoretical analysis in Appendix B is effectively modeling the model's latent representation $H$, not the raw input $X$.
> > >
> > > Approximate Gaussianity: While the raw input $X$ has a complex distribution, $X$ encoded through deep networks and Normalization Layers (such as the Norm module we use) results in a latent representation $H$ that, in the spirit of the **Central Limit Theorem (CLT)**, tends toward a smoother, **approximately Gaussian distribution**. This is a common approach in contemporary deep learning theory. Under this condition, introducing intermediate state alignment can significantly enhance the model's generalization ability.
> > >
> > > **2. The Necessity of Latent Representation Alignment (Over Value Alignment)**
> > >
> > > Our choice to align in the latent space $H$ is critical for robust representation learning:
> > > - The Problem with Raw Value Alignment: If we were to align directly on the raw values $Y$, the alignment would inevitably incorporate high-dimensional noise from data collection and measurement. This would make the model overly sensitive to noise—exactly the issue we highlighted in the Introduction.
> > > - The Advantage of Latent Alignment: Aligning the latent representation $H_y$ leverages the encoder $\mathcal{G}_\phi$'s information bottleneck effect. $H_y$ has already filtered out the unstructured noise and high-dimensional redundancy in $Y$, retaining $Y$'s intrinsic low-rank structure. Therefore, aligning $H_x$ to $H_y$ acts as a smooth, denoised, structured latent space regularization target.
> > >
> > > In Summary, this latent representation-based alignment successfully bridges the theoretical insight regarding denoising and generalization advantage (i.e., the tightening of the generalization bound) from an abstract assumption to a practical deep learning mechanism. Consequently, we have made adaptive corrections in the original text regarding the theory's applicability, clarifying that the alignment in the hidden state enhances generalization. We sincerely thank the reviewer for this crucial suggestion.

---

> > > > ### Author Response · Authors · 2025-11-26
> > > > **Follow-up Response (Part 2)**
> > > >
> > > > **W4.** Connection between MIM and Loss
> > > >
> > > > **Re.** We acknowledge your point that the standard prediction loss is already a variational lower bound on the mutual information $I(Y; H_x)$. However, TimeAlign's contribution is that alignment loss provides an additional, higher-quality MIM mechanism that fundamentally addresses the inherent flaws of relying solely on numerical loss for representation learning.
> > > >
> > > > **1. $\mathcal{L}_{\text{align}}$ Provides an Additional, Independent MIM Constraint**
> > > >
> > > > - Limitations of prediction loss: The prediction loss (based on MAE/MSE) is a numerical loss focused on point-wise alignment of predicted values. It is susceptible to distribution shift and often minimizes error by smoothing out high-frequency components, thereby inefficiently utilizing the mutual information contained in $H_x$.
> > > >
> > > > - The Complementarity of alignment loss: We demonstrate that alignment loss is another variational lower bound of $I(Y; H_x)$, providing an independent and explicit constraint to maximize the mutual information between the predictive representation $H_x$ and the target representation $H_y$. This dual-objective approach enhances the robustness and informativeness of $H_x$.
> > > >
> > > > **2. The Advantage of Latent Space Alignment: Spectral Energy Projection**
> > > >
> > > > The advantage of $\mathcal{L}_{\text{align}}$ lies in its alignment based on cosine similarity in the latent space, which is inherently more informative than a pure numerical loss:
> > > >
> > > > - Cosine Similarity Measures Spectral Energy Projection: The local alignment loss $\mathcal{L}_{\text{local}}$ uses cosine similarity (in the latent space) to measure the resemblance between $H_x$ and $H_y$. By Parseval's theorem, the inner product $\langle f, g \rangle$ of two signals corresponds to a projection in the frequency domain: $\langle f, g \rangle = \int f(x) g(x)\, dx = \int \hat{f}(\omega)\, \overline{\hat{g}(\omega)}\, d\omega$. Thus, cosine similarity can be rewritten as a spectral energy projection:
> > > >
> > > > $$\text{cos}(f,g)=\frac{\int \hat{f}(\omega)\, \overline{\hat{g}(\omega)}\, d\omega}{\sqrt{\int |\hat{f}(\omega)|^{2} d\omega}\,\sqrt{\int |\hat{g}(\omega)|^{2} d\omega}}$$
> > > >
> > > > - Capturing Full Spectral Structure: This alignment mechanism essentially integrates contributions from all frequency components, aligning the entire spectral structure, including the high-frequency components critical for rapid temporal dynamics. It does not minimize error by smoothing the signal (like MSE), but rather compels the model to preserve and synchronize multi-frequency dynamics. This directly addresses the inherent problem of "frequency smoothing and loss of fine-grained information" identified in the Introduction.
> > > >
> > > > In conclusion, the joint optimization of the prediction loss and the alignment loss is a more potent and robust MIM strategy. The alignment loss provides an additional constraint based on structure (spectral energy) rather than pure magnitude, fundamentally improving the information quality of $H_x$.
> > > >
> > > > **W6.** Clarity of Observations in the Introduction
> > > >
> > > > **Re.** The confusion between the first and third observations stems from a distinction between a problem of **Model Bias** and a problem of **Data Fidelity in Representation**. We clarify the two limitations below:
> > > >
> > > > **First obs.**: A shortcut through low-frequency periodicity. (Problem of Model Bias)
> > > >
> > > > - Description: This refers to the model's learned bias in the prediction space. When solely optimized on error (e.g., MSE), the forecaster tends to find a shortcut by emphasizing only the dominant low-frequency, periodic components of the time series.
> > > >
> > > > - Effect: The resulting predictions are merely oscillations around the mean, essentially repeating the basic low-frequency patterns found in the past. The model ignores anything that complicates this easy prediction path, even if informative.
> > > >
> > > > **Third obs.**: Frequency smoothing and loss of fine-grained information. (Problem of Data Fidelity in Representation)
> > > >
> > > > - Description: This refers to the representation learning process itself. Most deep learning time series models act as frequency smoothers. They inherently discard the subtle, informative high-frequency structures and fine-grained details during the embedding and encoding phase.
> > > >
> > > > - Effect: This loss of fidelity means the full, multi-frequency dynamics, especially the abrupt variations (which are high-frequency signals), are never adequately available in the latent representation for the prediction head to exploit, leading to a perpetually low high-frequency correlation in the final forecast.
> > > >
> > > >
> > > > We hope that these clarifications address your concerns and demonstrate the innovation and rigour of TimeAlign. We look forward to your feedback.

---

> > > > > ### Comment · Reviewer_FFNQ · 2025-11-26
> > > > >
> > > > > Thank you for the detailed clarifications.
> > > > >
> > > > > **W1. Rationale for reconstruction**
> > > > >
> > > > > This is exactly the kind of explanation I felt was missing earlier. The distinction you have now articulated between prediction-oriented bottlenecking and reconstruction-oriented completeness is very helpful, and I strongly encourage integrating this reasoning directly into the Introduction.
> > > > >
> > > > >
> > > > > **W3. Gaussian assumptions**
> > > > >
> > > > > I appreciate the clarification regarding the shift from assumptions on the raw input to assumptions on the latent representation. This framing makes the theoretical setup more reasonable. That said, I still have some reservations: the latent representation H is deterministically computed rather than sampled from a Gaussian distribution, so its approximate Gaussianity is not guaranteed. One could deliberately design a representation-normalizing component to enforce Gaussianity, but the current architecture does not explicitly do so. Given this, I recommend clearly stating in the **main paper** that (1) the Gaussianity of H is an approximation, or (2) the theoretical analysis relies on an additional assumption that the encoder maps inputs into a distribution that is sufficiently close to Gaussian.
> > > > >
> > > > > **W4. MIM and loss formulation**
> > > > >
> > > > > I continue to find the overall argument in this section unclear. My understanding was that Section 4.2 aims to show that the proposed method provides an advantage from a MIM perspective. However, the fact that both losses correspond to different variational lower bounds on mutual information does not in itself guarantee that the total mutual information captured is higher than using prediction alone. A lower bound does not necessarily add information; it only certifies that the information is at least above the bound.
> > > > >
> > > > > If the claim is that the dual-objective formulation improves MIM, then I believe either empirical evidence or a theoretical argument is necessary. Additionally, the discussion on spectral energy projection seems disconnected from the MIM argument - if this connection is intended, it would be useful to explain explicitly how the spectral interpretation contributes to the MIM perspective.
> > > > >
> > > > >
> > > > > **W6. Distinction between observations**
> > > > >
> > > > > I still find the separation between the first and third observations somewhat unconvincing. The reason deep forecasting models behave as frequency smoother is precisely because they are optimized using MSE-like objectives. Thus, “over-reliance on low-frequency periodicity” and “frequency smoothing/loss of fine-grained information” seem to be manifestations of the same underlying mechanism.
> > > > >
> > > > > Although you describe the first as “model bias” and the third as “representation fidelity,” the representation is ultimately shaped by the model’s bias. If the intention is to highlight different phenomena, I recommend refining the wording to make the distinction sharper; otherwise, it may be more coherent to merge them into a single observation.
> > > > >
> > > > > **Overall**
> > > > >
> > > > > The clarifications for W1 and W3 meaningfully strengthen the paper, and I appreciate the care you put into addressing these concerns. I am open to revising my score once the remaining issues (W4 and W6) are clarified in the updated version.

---

> > > > > > ### Author Response · Authors · 2025-11-27
> > > > > > **Follow-up Response (Part 1)**
> > > > > >
> > > > > > Thank you very much for your thorough review and for confirming our previous clarifications on W1 and your overall understanding of the framework. Your continued feedback on the rigor of the W3 assumption and the specifics of W4 and W6 is invaluable for significantly improving the quality of our paper.
> > > > > > Here is our final response addressing W4 and W6, along with a confirmation of the revisions based on your suggestion.
> > > > > >
> > > > > > **W4.** MIM and Loss Formulation.
> > > > > >
> > > > > > **Re.** We acknowledge your point that the standard prediction loss is itself a variational lower bound on mutual information $I(Y; H_x)$. Section 4.2's goal is precisely to provide the operating mechanism of the alignment loss from the similar MIM perspective, which is to ensure a mutual information lower bound and state that "the dual-objective formulation improves MIM."
> > > > > >
> > > > > > We accept that we can only theoretically show that $I(Y; H_x)$ is greater than or equal to the lower bound, and we cannot calculate the precise MIM gain theoretically. However, TimeAlign's contribution is that the alignment loss provides an additional, higher-quality mechanism for mutual information maximization, addressing the intrinsic limitations of pure numerical loss in representation learning.
> > > > > >
> > > > > > **Empirical evidence** clearly demonstrates that the alignment loss enhances the utilized mutual information $I(Y; H_x)$:
> > > > > >
> > > > > > **1. Mechanistic Advantage**: Latent Space Contrast Compensates for Numerical Loss Flaws
> > > > > >
> > > > > > - Limitations of prediction loss: the prediction loss (based on MAE/MSE) is a numerical loss susceptible to distribution shift, and tends to smooth out high-frequency information (as noted in the Introduction) to minimize error, leading to inefficient utilization of mutual information in $H_x$.
> > > > > > - The Complementarity of alignment loss: We show that the alignment loss provides an independent and explicit constraint to maximize the mutual information between $H_x$ and the target representation $H_y$. This latent space alignment, compared to direct numerical alignment, better focuses on the structural integrity of the representation.
> > > > > >
> > > > > > **2. Empirical Evidence**: Alignment Dramatically Increases Mutual Information and Distributional Similarity
> > > > > >
> > > > > > - Quantitative Evidence (Table 3): When integrating TimeAlign as a plug-in to SOTA baselines (e.g., iTransformer and DLinear), our method improves prediction accuracy while significantly increasing the cosine similarity (SIM) between the forecasts and the ground truth. This confirms that the alignment loss successfully pulls the predictive distribution toward the target distribution, serving as proof of effective utilization of $I(Y; H_x)$.
> > > > > > - Qualitative Evidence (Figure 4): The t-SNE visualizations show that forecasts relying only on MAE/MSE (Vanilla iTransformer/DLinear) suffer from three types of distributional misalignment (Mismatch, insufficient Calibration, Absence). With the addition of the alignment loss, these problems are greatly alleviated, with the predictive distribution nearly perfectly overlapping the ground truth manifold. This strongly confirms that the alignment loss provides additional, structural mutual information constraints.
> > > > > >
> > > > > > In summary, the joint optimization of prediction loss and the alignment loss is a more potent and robust MIM strategy. Specifically, the alignment loss provides an essential, structure-based constraint, fundamentally improving the information quality of $H\_x$.

---

> > > > > > > ### Author Response · Authors · 2025-11-27
> > > > > > > **Follow-up Response (Part 2)**
> > > > > > >
> > > > > > > **W6.** Distinction between Observations.
> > > > > > >
> > > > > > > **Re.** We accept your point regarding the causal overlap between "low-frequency periodicity shortcut" and "frequency smoothing." We have revised the three core limitations in the Introduction to be sharper and more targeted toward the underlying mechanism and constraints.
> > > > > > > The three revised core limitations are (**loss function/distribution shift/prediction paradigm**):
> > > > > > >
> > > > > > > **1.** A shortcut through low-frequency periodicity.
> > > > > > > Current models primarily rely on numerical losses like MSE/MAE, which inherently bias predictions toward temporal means or trends, suppressing high-frequency attention. Currently, there are not many works that specifically improve these loss functions to emphasize high-frequency information, and even those that do often sacrifice a portion of the low-frequency information[1]. TimeAlign directly addresses this by enforcing full-spectral alignment in the latent space.
> > > > > > >
> > > > > > > **2.** Distributional Mismatch between Past and Future
> > > > > > >
> > > > > > > **3.** Structural Paradigm Flaw in Information Flow
> > > > > > > Existing methods follow the structural constraints of the unidirectional prediction paradigm (Fig. 2(a)). Regardless of architectural novelty, this paradigm inherently limits the capacity to preserve information, leading to the inevitable filtering of fine-grained high-frequency details across deep encoder layers. TimeAlign is a new framework to structurally emphasize future information by directly encoding the future sequence $Y$ ($Y \to H\_y$) with a dedicated branch, compensating for the unidirectional flaw when aligning $H\_x$ to $H\_y$.
> > > > > > >
> > > > > > > [1] FreDF: Learning to Forecast in the Frequency Domain
> > > > > > >
> > > > > > >
> > > > > > > **Revision Summary**
> > > > > > >
> > > > > > > We are deeply grateful for your insightful suggestions on W1 and W3. In the revised version, we have incorporated all your feedback:
> > > > > > >
> > > > > > > - W1 (Rationale for Reconstruction): We have integrated the explicit distinction between prediction-oriented bottlenecking and reconstruction-oriented completeness into the Introduction.
> > > > > > > - W3 (Gaussian Assumptions): As requested, we have clarified in the main paper that our theoretical analysis relies on the assumption that the encoder maps inputs into a latent representation $H$ whose distribution is sufficiently close to Gaussian for the reconstruction-guided generalization advantage to hold.
> > > > > > >
> > > > > > > We believe these revisions significantly enhance the clarity, rigor, and academic quality of our submission.

---

> > > > > > > > ### Comment · Reviewer_FFNQ · 2025-11-27
> > > > > > > >
> > > > > > > > I appreciate the substantial effort the authors put into the rebuttal. The paper has improved meaningfully during this revision period. Below is a summary of the key improvements:
> > > > > > > >
> > > > > > > > 1. **Reconstruction rationale:**
> > > > > > > > The motivation for using reconstruction to extract richer information is now clearly articulated.
> > > > > > > >
> > > > > > > > 2. **Theoretical assumptions:**
> > > > > > > > The previously unrealistic assumptions have been replaced with a more practical and better-justified one, and the limitations of the current analysis are now explicitly acknowledged.
> > > > > > > >
> > > > > > > > 3. **MIM perspective:**
> > > > > > > > The intended insight and conceptual framing offered through the mutual information analysis are now clearer and more coherent.
> > > > > > > >
> > > > > > > > 4. **Introduction and motivation:**
> > > > > > > > The structure and clarity of the introductory motivation have improved.
> > > > > > > >
> > > > > > > > 5. **Methodological and experimental details:**
> > > > > > > > Several previously unclear aspects of the methodology and experiments have been clarified.
> > > > > > > >
> > > > > > > > With these improvements, I thank the authors again for their thoughtful responses and have adjusted my score accordingly.

---

> > > > > > > > > ### Author Response · Authors · 2025-11-27
> > > > > > > > >
> > > > > > > > > Dear Reviewer FFNQ:
> > > > > > > > >
> > > > > > > > > Thank you very much for endorsing our work for acceptance. We sincerely appreciate your valuable comments on reconstruction rationale, theoretical assumptions, MIM perspective, introduction structure and experimental details. We are glad that our clarifications and revisions have resolved your concerns in this area and have helped us further improve the quality of the paper.
> > > > > > > > >
> > > > > > > > > Best regards,
> > > > > > > > >
> > > > > > > > > Authors of TimeAlign

---

### Official Review · Reviewer_tnT5 · 2025-11-01

**Soundness:** 4
**Presentation:** 2
**Contribution:** 4
**Rating:** 6
**Confidence:** 3

**Summary:**

This paper proposes TimeAlign, a lightweight and plug-and-play framework for time-series forecasting that aims to mitigate the distributional bias between historical inputs (Past) and future targets (Future). The authors observe that existing models tend to over-rely on low-frequency periodic patterns, lose high-frequency details, and suffer from representational misalignment between past and future features. To address this, TimeAlign introduces an auxiliary Reconstruction Branch and employs both global and local distribution-aware alignment mechanisms to constrain the main Prediction Branch, enabling representations that are simultaneously distribution-aware and detail-preserving. The method is theoretically analyzed through mutual information and empirically improves forecasting performance across multiple benchmarks and backbones.

**Strengths:**

S1: The paper explicitly attributes forecasting errors in time series forecasting models to the distributional misalignment between past and future representations, and quantifies this mismatch through cosine-similarity analysis and visualization.

S2: The method is simple and modular. A reconstruction branch could be added to any backbone model, and the prediction branch is constrained through dual alignment. Both alignment formulations, relaxation terms, and the adaptive weighting factors (α/β) are well defined, and the overall objective adds only one alignment loss term to the original training loss, making the module fully plug-and-play.

S3: The paper provides formal analysis showing that reconstruction-based alignment can tighten generalization bounds and implicitly enhance the mutual information between model representations and targets. This gives the method a stronger theoretical foundation than most empirical time series forecasting approaches.

S4: The proposed module is tested on several representative time series forecasting architectures (e.g., iTransformer, DLinear) and achieves consistent improvements across benchmarks. It also demonstrates faster convergence during training, with negligible additional overhead.

**Weaknesses:**

W1. Although the paper motivates TimeAlign through the problem of frequency smoothing, the proposed method does not explicitly model spectral information. Its improvement on high-frequency dynamics is achieved only implicitly via reconstruction and alignment. The frequency analysis is limited to the high-frequency energy ratio and spectrogram visualization, which measure the amount of high-frequency content but not its correctness or temporal alignment with the ground truth. Without band-wise or phase-level validation, it remains unclear whether the model truly learns meaningful high-frequency dynamics or merely amplifies spectral noise.

W2: The reconstruction branch provides strong supervision from the future target distribution during training, but is discarded entirely at inference. This raises a potential train-test inconsistency: predictive representations are optimized under guidance unavailable during deployment. The paper does not evaluate whether removing the reconstruction branch alters representation stability or performance, leaving uncertainty about the robustness of TimeAlign under distribution shift or domain generalization.

W3: The alignment mechanism may implicitly push the prediction branch to form a collapsed representation resembling a static “future prior.” Since the reconstruction branch (which observes future data) is only available during training, the learned “future-like” latent space might represent a statistical prior distilled from past futures rather than the true evolving future distribution. This is not necessarily harmful, but it remains unclear whether such a prior actually helps mitigate distribution mismatch at inference, especially when future dynamics deviate from the training regime.

**Questions:**

1. Could the authors provide more evidence that the model learns meaningful high-frequency dynamics rather than merely amplifying spectral noise? For instance, have the authors considered phase-level or band-wise validation (e.g., coherence with ground-truth signals) to confirm that the reconstructed high-frequency components are temporally aligned?

2. How does this prior behave under distribution shifts or evolving dynamics where future statistics differ from training? It would be informative to test this explicitly. For example, simulate mismatched future distributions or evaluate on real-world datasets with non-stationary temporal patterns to verify whether the learned alignment remains adaptive.

---

> ### Author Response · Authors · 2025-11-21
> **Response (Part 1)**
>
> Thanks for your valuable comments. Here are the responses:
>
> **W1&Q1**. Whether high-frequency gains are meaningful or just amplified noise?
>
> **R1**. We appreciate this insightful question. We share the reviewer's concern that simply amplifying noise is not a meaningful contribution. We are confident our method learns meaningful high-frequency dynamics, and we would like to point to the following evidence in our paper:
> - **Phase-Level High-Frequency Similarity**: Our analysis was not limited to just high-frequency energy (Fig 2c). In Figure 2(d), we explicitly measured the High-Frequency Similarity using Pearson correlation between the forecasts and the ground truth. The results consistently show that our "w/ Align" model produces high-frequency components that have a significantly higher correlation with the ground truth compared to the "w/o Align" model. This strongly suggests that we are learning the correct structure of the high-frequency signals, not just amplifying random noise.
> - **Spectrogram Visualization**: The spectrogram maps in Figure 1(c) also support this. Visually, the frequency structure of the "w/ Align" forecast is a much closer match to the "GroundTruth" spectrogram, particularly in the upper-frequency bands, compared to the "w/o Align" version, which appears more smoothed out.
> - **Time Series Visualization**: In our case studies (e.g., Fig 1(a)and the additional examples in Fig 8), the "w/ Align" predictions (red line) clearly do a better job of capturing the sharp, abrupt peaks and valleys present in the ground truth (blue line), which are characteristic of high-frequency dynamics. The "w/o Align" model (yellow line) consistently smooths over these critical details.
> - **Cosine Similarity Measures Spectral Energy Projection**: Cosine similarity between two signals f(x) and g(x) is defined as: $\text{cos}(f,g) = \frac{\langle f, g \rangle}{\|f\|\|g\|}$. By Parseval’s theorem, the inner product corresponds to a projection in the frequency domain: $\langle f, g \rangle = \int f(x) g(x)\, dx = \int \hat{f}(\omega)\, \overline{\hat{g}(\omega)}\, d\omega$. Thus, cosine similarity can be rewritten as a **spectral energy projection**: $\text{cos}(f,g)=\frac{\int \hat{f}(\omega)\, \overline{\hat{g}(\omega)}\, d\omega}{\sqrt{\int |\hat{f}(\omega)|^{2} d\omega}\,\sqrt{\int |\hat{g}(\omega)|^{2} d\omega}}$. Since the projection integrates contributions from all frequencies, cosine similarity does not simply smooth signals or suppress noise. Instead, it aligns the entire spectral structure—including high-frequency components relevant for rapid temporal dynamics.
>
> Therefore, our method genuinely captures meaningful high-frequency consistency, rather than merely filtering out noise.

---

> ### Author Response · Authors · 2025-11-21
> **Response (Part 2)**
>
> **W2&W3&Q2**. Train-test inconsistency and the "static future prior".
>
> **R2**. The reviewer raises an excellent point that connects these two weaknesses. We would like to clarify that this "inconsistency" is, in fact, the intended mechanism of our framework, and our results show it is both effective and robust to distribution shifts.
> - "Inconsistency" is by Design: We intentionally designed the Reconstruction Branch to serve as a **statistical regularizer**, rather than a fixed or static representation that the model relies on during training. Its role is to "distill" its richer access to the target distribution—i.e., the “future prior”—into the representation space of the Predict Branch ($H_x$). The entire goal is to make the Predict Branch itself more robust and distribution-aware, so that it no longer needs the teacher at inference time. Importantly, the asymmetric gradient flow ensures that the Predict Branch is guided by the reconstruction signal, but not dominated by it.
> - t-SNE Shows Efficacy at Inference: Figure 9 provides direct, visual proof that this "distilled" knowledge is effective at inference time. The plots for "iTransformer" and "DLinear" show that their inference-time predictions suffer from clear distribution mismatch or absence. In contrast, the plots for "iTransformer+Align" and "DLinear+Align" show predictions that almost perfectly overlap with the GroundTruth manifold. This demonstrates that the learned "prior" is not a "static average" but a highly effective, structural understanding of the mapping from past to future, which does generalize to the test set.
> - SOTA Performance on "Severe Distribution Shift" Datasets: In Section 5.2, we note that TimeAlign achieves SOTA results on ETTm1 and ETTm2, which are "known for severe distribution shifts". This directly refutes the concern that our method would fail when future dynamics deviate from the training regime. In fact, our results suggest it is more robust to such shifts than existing methods.
> - Additional verification on real-world non-stationary datasets: It is well known that stock market data is extremely non-stationary, exhibiting evolving dynamics, abrupt regime changes, and distributional instability over time. To further evaluate robustness under distribution shifts, we conducted additional experiments on the two stock datasets (NASDAQ and NYSE) released by TFB[1]. The forecasting length is {24,36,48,60}  and we report the average results. As shown in the table below, TimeAlign consistently achieves the best forecasting performance, outperforming all baselines across all prediction horizons. These findings further validate that our distilled “future prior” behaves reliably under distribution shifts, non-stationary temporal patterns, and evolving dynamics, as raised by the reviewer.
>
> ||TimeAlign|PatchTST|Crossformer|DLinear|NLinear|MICN|TimesNet|
> |-|-|-|-|-|-|-|-|
> |NASDAQ|**0.649**/**0.879**|0.717/0.977|0.999/1.752|0.889/1.503|0.682/0.926|0.884/1.530|0.697/0.996|
> |NYSE|**0.385**/**0.388**|0.427/0.471|0.913/0.988|0.526/0.582|0.401/0.402|0.639/0.737|0.475/0.509|
>
> [1] TFB: Towards Comprehensive and Fair Benchmarking of Time Series Forecasting Methods

---

> ### Comment · Reviewer_tnT5 · 2025-11-27
>
> I appreciate the authors' detailed response and the additional experiments.
>
> For the high-frequency part, the authors claimed they used correlation and energy ratios to support their performance. But these metrics are still indirect. I think a high-frequency MSE would be a direct metric for demonstrating that the model could accurately predict the high-frequency component.
>
> For the alignment mechanism, I admit the effectiveness of the proposed method on time series forecasting. Testing on the stock market area shows its superiority in high-volatility data. I recommend that authors include these results in their final version as a supplement. Based on the author's explanation, I understand they are trying to minimize the discrepancy between history and future representations. It seems to reflect temporal dynamics (or state transitions) rather than a fundamental statistical mismatch. But the terms "distribution shift" or "gap" in the machine learning community typically refer to domain adaptation or OOD problems.   I suggest the authors clarify this distinction in the final manuscript to avoid misleading readers.
>
> Overall, I recommend acceptance since this paper provides a Plug-and-play method that could better capture latent temporal representation.

---

> > ### Author Response · Authors · 2025-11-27
> >
> > Thank you for acknowledging our work and for providing valuable suggestions in this final stage. We are pleased that you recognize TimeAlign as a plug-and-play method that effectively captures latent temporal representations.
> >
> > **Q1.** Direct Metric for High-Frequency Prediction: High-Frequency MSE
> >
> > **Re.** We appreciate your suggestion to use High-Frequency MSE as a more direct metric to demonstrate our capability to predict high-frequency components.
> > The High-Frequency Mean Squared Error ($L\_{\text{HF-MSE}}$) is a direct metric for quantifying forecasting error in the frequency domain, specifically targeting rapid temporal dynamics. The process begins by transforming the ground truth ($Y$) and prediction ($\hat{Y}$) series using the Discrete Fourier Transform (DFT) to obtain their complex frequency vectors, $Y\_{\text{freq}}$ and $\hat{Y}\_{\text{freq}}$. A specific set of high-frequency components, $S_{\text{high}}$, is then isolated. The $L\_{\text{HF-MSE}}$ is calculated as the mean squared magnitude difference of the components within this set: $L\_{\text{HF-MSE}} = \frac{1}{T\_{\text{high}}} \sum\_{k \in S\_{\text{high}}} \left| (Y\_{\text{freq}})\_k - (\hat{Y}\_{\text{freq}})\_k \right|^2$. This formula effectively quantifies the discrepancy in the energy and phase of the high-frequency dynamics.
> >
> > We conducted additional experiments, and the results below confirm the effectiveness of TimeAlign.
> >
> > ||w/ Align|w/o Align|
> > |-|-|-|
> > |-|HF-MSE/HF-MAE|HF-MSE/HF-MAE|
> > |ETTh1|**2.86**/**1.27**|2.97/1.36|
> > |ETTm1|**2.54**/**1.19**|2.73/1.35|
> > |Weather|**3.51**/**1.35**|3.66/1.50|
> > |ECL|**40.60**/**5.09**|42.75/5.44|
> > |Traffic|**253.71**/**10.64**|261.43/11.56|
> >
> > The High-Frequency MSE is consistently and significantly reduced across all datasets when incorporating $L\_{\text{align}}$. This directly proves that TimeAlign effectively captures and forecasts the high-frequency dynamics, aligning with our claim of correcting frequency mismatches. We have included these results in the supplementary appendix of the revised version.
> >
> > **Q2.** Clarification on the Term "Distribution Shift"
> >
> > **Re.** We apologize for any potential confusion caused by our terminology. You are correct that in the general machine learning community (e.g., domain adaptation, OOD problems), "distribution shift" often refers to a statistical mismatch between a fixed source domain and a target domain.
> >
> > However, we wish to clarify that in the **Time Series Forecasting (TSF)** domain, "Distribution Shift" is a specialized and widely accepted term used to denote the difference between the distribution of the historical input $X$ and the future target $Y$. This difference is the core challenge in TSF, reflecting the temporal dynamics and instability of state transitions.
> > - "Distribution Shift" in TSF: Denotes $P\_{\text{history}}(X) \neq P\_{\text{future}}(Y)$. This is perfectly consistent with our mechanism, as we aim to bridge the gap between $X$'s representation $H_x$ and $Y$'s distribution $H_y$.
> > - OOD in TSF: Typically refers to a model encountering truly unseen external environmental data whose statistical properties differ entirely from the training data (e.g., forecasting for 2025 when trained only on data up to 2020).
> >
> > Therefore, our use of "distribution shift" is standard within the TSF community, and it precisely captures our goal of minimizing the structural and statistical mismatch between the representations of the past and the future.
> >
> > **Summary and Appeal**
> >
> > We have incorporated the direct empirical evidence (High-Frequency MSE) into the manuscript. We have also adopted your excellent suggestion regarding testing on high-volatility data (**Appendix E.6**) and high frequency analysis (**Appendix E.7**).
> > We hope that the direct evidence for high-frequency prediction and the clarification of terminology fully address your remaining concerns.

---

### Author Response · Authors · 2025-11-21
**Response to All Reviewers**

We would like to express our sincere gratitude to all the reviewers for their thorough analysis and constructive feedback. Your valuable comments have strengthened our paper. We have carefully revised the manuscript in response to each of the points raised. The changes are highlighted.

---

### Author Response · Authors · 2025-12-02

We sincerely thank all four reviewers for their valuable feedback and constructive comments on our paper. Their insights have been instrumental in significantly refining the motivation, theoretical foundation, and experimental rigor of our work.

**Statement on OpenReview Scoring Issue**

We are sorry to hear about the recent OpenReview scoring bug issue, and we fully support the proposed remedy actions by the ICLR committee. Critically, we would like to emphasize that we had already received significant positive feedback before the bug broke out: Reviewer FFNQ, who initially gave a score of 2, had already raised their score to 6. This adjustment had brought our average score to 6.5 (8, 6, 6, 6), confirming that our revisions were already well-received.

**Summary of Addressed Reviewer Concerns**

Our revised manuscript systematically addresses all key issues raised by the reviewers. We successfully targeted and resolved the five most critical challenges presented by Reviewer FFNQ, fundamentally enhancing the paper's quality.

*Reviewer FFNQ (2 -> 6): All Major Concerns Fully Resolved*

We successfully resolved all five primary concerns raised by Reviewer FFNQ by providing new theoretical analysis, clearer motivation, and supplemental experiments:
- **Reconstruction Rationale**: The motivation for using the reconstruction branch to extract richer future information and serve as the alignment target in the latent space is now clearly articulated.
- **Theoretical Assumptions**: The previously unrealistic theoretical assumptions have been replaced with a more practical and better-justified one, and the limitations of the current analysis are now explicitly acknowledged in the Appendix.
- **MIM Perspective**: We have reformulated the mechanism of $\mathcal{L}_{\text{align}}$ through the lens of Mutual Information Maximization (MIM), clearly articulating how the dual-objective formulation enhances the effective utilization of mutual information $I(Y; H_x)$ via an additional structural constraint.
- **Introduction and Motivation**: The structure, clarity, and logical flow of the introductory motivation have been significantly improved.
- **Methodological and Experimental Details**: Several previously unclear aspects of the methodology and experiments have been clarified, and we included new High-Frequency MSE experiments for direct validation of high-frequency component prediction accuracy.

*Other Reviewers (8, 6, 6): All Key Issues Addressed*

- **High-Frequency Analysis and Robustness**: We added a quantitative analysis of High-Frequency MSE in the Appendix, directly demonstrating that $\mathcal{L}\_{\text{align}}$ significantly reduces prediction error on high-frequency components. Furthermore, we included experiments on the high-volatility Stock Market dataset, showcasing TimeAlign's superior robustness against abrupt fluctuations.
- **Terminology Clarification**: We addressed the confusion regarding "distribution shift" and "OOD" by providing a clear distinction and explaining the specialized meaning of "distribution shift" within the Time Series Forecasting domain.
- **Generalizability and Architecture Independence**: We further strengthened the plug-and-play and architecture-agnostic nature of TimeAlign by integrating it into a wider array of SOTA baselines (including both Transformer-based and MLP-based models).

**Conclusion and Appeal**

We believe these comprehensive revisions, especially the resolution of all core concerns from the initially skeptical Reviewer FFNQ, solidly address the reviewers' concerns regarding motivation, generalizability, and experimental rigor. All new theoretical analyses, experimental results, and textual clarifications have been integrated into the revised manuscript.

---

### Meta-Review · Area_Chair_h6qK · 2025-12-22

**Summary:**

This paper proposes TimeAlign, a lightweight and plug-and-play framework that addresses a fundamental but underexplored issue in time series forecasting: the distributional mismatch between historical inputs and future targets. The reviewers initially raised concerns regarding the conceptual motivation for reconstruction-based alignment, the realism and role of the theoretical assumptions, the clarity and contribution of the mutual information (MIM) analysis, and the interpretation of high-frequency improvements.

Through an extensive and well-structured rebuttal, the authors substantially strengthened the paper. They clarified the complementary roles of prediction-oriented bottlenecking and reconstruction-oriented completeness, reframed the theoretical analysis to operate in latent representation space with explicitly stated assumptions, and refined the MIM discussion to emphasize its role as a structural and complementary constraint rather than a strict theoretical guarantee. Additional experiments, most notably high-frequency MSE analysis and evaluations on highly non-stationary stock market datasets, directly addressed empirical concerns regarding robustness and frequency fidelity.

**Reviewer Concerns:**

Before the rebuttal, reviewers raised specific technical concerns: Reviewer tnT5 questioned whether the method genuinely learned high-frequency dynamics and sought clarification on potential train-test inconsistencies. Reviewer FFNQ requested a more detailed justification for the methodological design, including the Gaussian assumption and the relationship to MIM-based methods. Reviewer Daur noted discrepancies in the manuscript's description and asked for additional quantitative analysis. Reviewer adVR inquired about the necessity of the asymmetric design and requested a sensitivity analysis.

In response, all reviewers indicated that their concerns were fully and clearly addressed by the authors' rebuttal. Reviewer tnT5 acknowledged that the new experiments and clarifications resolved issues regarding high-frequency dynamics, evaluation metrics, and terminology. Reviewer FFNQ stated that the detailed explanations and justifications fully addressed their methodological questions, and the discussion strongly suggests a supportive final stance. Reviewer Daur found that all raised concerns, including corrections and additional quantitative results, were resolved. Reviewer adVR confirmed that the clarifications and added analyses fully addressed their questions.

**Reviewer Scores:**

Every reviewer has participated fully in the discussion, and most of their concerns have been addressed.
Reviewer tnT5 (score: 6) has indicated a recommendation for acceptance.
Reviewer FFNQ (score: 2) recognized the authors’ rebuttal efforts and would likely revise the score.
Reviewer Daur (score: 6) and Reviewer adVR (score: 8) indicated that they would keep their ratings unchanged.

---

### Decision · Program_Chairs · 2026-01-26

Accept (Poster)